# Review of Venoms of Non-Polydnavirus Carrying Ichneumonoid Wasps

**DOI:** 10.3390/biology10010050

**Published:** 2021-01-12

**Authors:** Donald L. J. Quicke, Buntika A. Butcher

**Affiliations:** 1Integrative Ecology Laboratory, Department of Biology, Faculty of Science, Chulalongkorn University, Phayathai Road, Pathumwan 10330, Thailand; d.quicke@email.com; 2Center of Excellence in Entomology, Bee Biology, Diversity of Insects and Mites, Chulalongkorn University, Phayathai Road, Pathumwan 10330, Thailand

**Keywords:** Ichneumonidae, Braconidae, *Pimpla*, *Aphidius*, *Habrobracon*, *Asobara*

## Abstract

**Simple Summary:**

Parasitoid wasps play an important role in all terrestrial ecosystems (both natural and agricultural) regulating the population densities of many herbivorous insects. Venom components have been studied in a number of exemplar species, both from a pure science perspective and because of their potential as lead structures for novel insecticides. The Ichneumonoidea comprises some 40,000 described species and several times more undescribed ones. We review the chemistry and physiological actions of these wasp venoms. Only those species that lack genome-encoded polydnaviruses are considered because of the complexity of interactions between venom and polydnavirus. Examples include ecto- and endoparasitoids, and idiobionts (host development arrested at time of parasitisation) and koinobionts (hosts allowed to continue developing). Most associations are with Lepidoptera. Unlike aculeate (stinging) wasps, bees and ants, venoms are dominated by proteins with only a few containing small peptides or biogenic amines. Venom effects include temporary, permanent or no host paralysis, host castration, immune suppression and modifying host metabolism. Few venom components have clearly identified effects. Studied venoms involve only a very small number of ichneumonoid subfamilies outside of those producing polydnaviruses. Suggestions are made regarding other systems that may be worthwhile investigating.

**Abstract:**

Parasitoids are predominantly insects that develop as larvae on or inside their host, also usually another insect, ultimately killing it after various periods of parasitism when both parasitoid larva and host are alive. The very large wasp superfamily Ichneumonoidea is composed of parasitoids of other insects and comprises a minimum of 100,000 species. The superfamily is dominated by two similarly sized families, Braconidae and Ichneumonidae, which are collectively divided into approximately 80 subfamilies. Of these, six have been shown to release DNA-containing virus-like particles, encoded within the wasp genome, classified in the virus family Polydnaviridae. Polydnaviruses infect and have profound effects on host physiology in conjunction with various venom and ovarial secretions, and have attracted an immense amount of research interest. Physiological interactions between the remaining ichneumonoids and their hosts result from adult venom gland secretions and in some cases, ovarian or larval secretions. Here we review the literature on the relatively few studies on the effects and chemistry of these ichneumonoid venoms and make suggestions for interesting future research areas. In particular, we highlight relatively or potentially easily culturable systems with features largely lacking in currently studied systems and whose study may lead to new insights into the roles of venom chemistry in host-parasitoid relationships as well as their evolution.

## 1. Introduction

Parasitoids are predominantly insects that develop as larvae on or inside their host, also usually another insect, ultimately killing it [1,2]. For various periods of time both parasitoid larva and host are alive even if the latter is moribund, and so during this period the relationship is parasitic, and interactions between host and parasitoid are mediated by venoms and sometimes other chemical secretions.

Among the Hymenoptera, the huge superfamily Ichneumonoidea comprising a minimum of 100,000 species, is composed almost entirely of parasitoids of other insects [3]. The extant Ichneumonoidea is dominated by two similarly sized families, the Braconidae and Ichneumonidae as well as and one small, recently recognised family, Trachypetidae [4]. The Braconidae and Ichneumonidae are collectively divided into approximately 80 subfamilies, most of which comprise species with a relatively uniform biology. Within each family there are parasitoids of Lepidoptera, Coleoptera, Diptera and other insect orders, there are ecto- and endoparasitoids, and there are representatives of two major life history strategies, the idiobionts and koinobionts [5]. Idiobionts curtail host development at the time of parasitisation and, in the case of ectoparasitoids, generally immobilise the host with a permanently paralysing venom. Koinobionts, on the other hand, allow the host to continue feeding and developing and often metamorphosing at the time of attack, and the female wasp’s venom typically has more subtle and longer acting effects on the host, though they may also induce temporary paralysis to facilitate oviposition. An abridged summary of the current state of knowledge of ichneumonoid phylogeny is presented in Figure 1, which also indicates a number of biological attributes. Estimates of ease of culturing and of suitability for transcriptomic studies are based on ease of rearing hosts, potential for multiple generations per year, body size and ease of capture at suitable localities and time of year—expert identification help should always be sought in such a large and taxonomically challenging group as the ichneumonoids.

An advantage of working with ectoparasitoids is that venom actions can be isolated from any possible parasitoid larva-induced effects simply by removing the externally deposited parasitoid eggs before they hatch. However, even with some endoparasitoids it is possible to disturb the wasp after initial envenomation and before an egg has been deposited in the host, e.g., [10].

Members of six ichneumonoid subfamilies have been shown to release DNA-containing, virus-like particles, encoded within the wasp genome, which are classified in the virus family Polydnaviridae [11]. These particles are called polydnaviruses (PDVs) but unlike true viruses they do not replicate, either within the wasp or in its host. They do, however, penetrate particular host cells where their genes are expressed resulting in profound effects upon host physiology. The effects of PDVs are complicated and often involve interactions with various venom and ovarial secretions, and they have attracted an immense amount of research interest [12,13,14,15,16,17,18,19]. PDVs are produced by cells of the calyx glands at the distal end of the lateral oviducts and not from the venom glands proper. They were first discovered in the braconid subfamily Cardiochilinae [20], though amazingly the first protective particles discovered were in the campoplegine ichneumonid, *Venturia canescens* [21,22,23], and these do not contain any viral DNA. Nevertheless, the particles produced by *Venturia* appear to have been derived from the true campoplegine polydnaviruses. Soon thereafter, a different group of polydnaviruses were found in other members of the Campopleginae (indeed all other investigated species). PDVs in the Braconidae appear to be ubiquitous in the ‘microgastroid’ group of subfamilies, but not elsewhere: they have been demonstrated in the Cardiochilinae, Cheloninae, Microgastrinae, and Miracinae and they are presumed also to occur in the relatively rare Dirrhopiinae, Khoikhoinae and Mendeselinae because of their phylogenetic relationships [5]. Much more recently, true polydnaviruses of a separate evolutionary origin were discovered in the ichneumonid subfamily Banchinae [17,18,24]. Because of the complexity of interactions of polydnaviruses, venom components [16], but see also [25] and probably often also with parasitoid larval secretions, e.g., [26], we do not consider any of these subfamilies further. In addition to PDVs, non-PDV viruses are a component of the venom gland (VG) secretions in some other braconids [27].

A few recent investigations have concerned the toxic effects of parasitoid venoms on cultured cell lines and in particular on whether they might contain compounds of potential therapeutic use against cancer cells [28,29]. Such activities would not be apparent from many studies that have concentrated on aspects such as host paralysis or melanisation.

To date there have been very few immunological studies of venoms from non-PDV ichneumonoids to investigate cross-reactivity between parasitoid species, and with the spectacular advent of proteo-transcriptomics it seems unlikely that there will be any future need. Here we review the literature on the relatively few studies on the effects and chemistry of these ichneumonoid venoms and also make suggestions for interesting future research areas. In particular, we highlight relatively easily culturable systems with features largely lacking in currently studied systems and whose study may lead to new insights into the roles of venom chemistry in host-parasitoid relationships.

### 1.1. Difficulties Working on Parasitoid Wasp Venoms

Before going into details about the actions of venoms and case studies, it is worth contemplating on some of the difficulties that are involved. Obviously many parasitoids are small so venom quantities are likely to be limiting. Many individual wasp females will be needed and in the case of parasitoids that essentially means having to keep hosts in culture in sufficient numbers to generate parasitoids for study. This actually results in a trade-off: the majority of the best studied parasitoid venoms involve small-bodied species that are relatively easily cultured in very large numbers: *Habrobracon hebetor* a gregarious parasitoid of grain moths, *Asobara tabida* which attacks *Drosophila* and *Aphidius ervi* which develops on various pest aphids. All are multivoltine and thus despite their physical size, can be reared so they are available on a daily basis. The only well-studied large parasitoids are pimplines which are polyphagous and easily cultured on greater wax moth larvae—indeed there has been some success in culturing *Pimpla* and other Pimplinae on artificial media rather than on living hosts [30,31,32,33,34,35,36]. Imagine the difficulty of studying the effects of obligate hyperparasitoids such as the mesochorine ichneumonids at the fourth trophic level. To do so would almost certainly have to involve wild collected individuals (a few species are associated with and more readily accessible from pest systems (e.g., *Mesochorus discitergus*, a parasitoid of the microgastrine *Cotesia marginiventris* on green cloverworm, *Plathypena scabra* [37]). However, these are likely to be far away from the laboratories of insect physiologists.

Another major aspect that may impact on the ability to maintain parasitoids in culture is seasonality. Parasitoids that are strongly seasonal, maybe monovoltine with monovoltine hosts and/or have to undergo obligate diapause, are necessarily going to be difficult to study. Even in the tropics there is often marked seasonality (wet and dry seasons) but most physiological research laboratories are not located in the tropics, they are in cool, temperate Europe, North America and Japan. It may nevertheless be possible to obtain sufficiently large numbers of individuals of some species during their active season and preserve (deep freeze) venom glands or venoms for study through the coming year.

### 1.2. The Disparity between Physiologists, Taxonomists and Natural Historians

It seems paradoxical that some of the best studied and observed larger parasitoid wasps have never been investigated from a venom point of view. A quick internet search for the often common genus, *Megarhyssa* (Ichneumonidae: Rhyssinae), very large wasps by any standard, will yield hundreds of hits but there has been only one, as yet unpublished, study (see Section 4.2.3). These parasitoids in North America and Palaearctic attack wood-boring siricid wood wasp larvae [38]. The adult females can often be seen ovipositing in considerable numbers on logs in wood piles. Such wasps are not easy to culture though this has been achieved by some pest control workers, but they are easy to collect during the summer. Their venom reservoirs are large, and it would not be difficult to collect wasps, dissect them and sore their venom reservoirs or extracted venom for research throughout the year. However, physiologists are seldom field naturalists or taxonomists, so collaboration is essential. For example, two ichneumonid wasps included in one study of Hymenoptera venom proteins were not even identified as far as subfamily despite one of the co-authors being a well-known hymenopterist [39]. Ichneumonoid identification can be challenging and needs appropriate expertise.

### 1.3. Non-Venom Sources of Chemical Interaction between Host and Parasitoid

In this review we only deal in detail with those wasps which do not possess genomically encoded polydnaviruses and inject these into their hosts along with their eggs and venom. The interactions between PDVs and venoms are complicated. However, VG products are not the only way in which parasitoids can modulate host physiology. From the adult female wasp there may also be secretions originating from the ovaries and oviducts. There is evidence that larval secretions both from the mouth and anus may affect hosts [40,41], and several groups of braconids produce teratocytes, which are giant cells formed from dissociated cells of their extra-embryonic membranes, and these can also secrete physiologically active and antimicrobial compounds into the host [42,43,44,45]. Many of the polydnavirus-containing parasitoid groups also form teratocytes, but so too do members of the Aphidiinae and Euphorinae [5,46]. Therefore, investigating the roles of pure venom compounds becomes rather tricky.

### 1.4. Transcriptome Studies

Advances in genomics and RNA sequencing technology and in mass spectrometry of proteins are revolutionising the investigation parasitoid wasp venoms [47], which hopefully will enable far more species to be investigated from a comparative point of view.

Total VG, reservoir and duct RNA or just VG transcriptomes have been sequenced for several ichneumonoid species, and also venom proteins have been separated and identified using LC-MS-MS for others (Table 1).

### 1.5. Taxonomic Notes

Physiologists are not normally that up to date with the often complex taxonomic changes that take place as a result of the increased understanding of phylogenetic relationships. Therefore, many of the genus and species names referred to in the venom (and other physiological) literature are now incorrect. In Table 2, we provide a summary of the correct names for the wasps we refer to. Other potentially relevant members of the braconid subfamily Opiinae, the correct combinations have been summarised [5,59].

## 2. Implications Based on Venom Apparatus Morphology

The venom glands (VGs), associated chitin-lined venom reservoir and primary venom duct, which together comprise the venom apparatus, show enormous variation in both braconids [2,60,61,62,63,64] and ichneumonids [6,65]. Edson and Vinson [66] classified braconid VG reservoirs into Type I which have thick musculature and well-developed spiral ridges (Figure 2C–E) as opposed to Type II venom apparatus which have reservoirs with thinner walls and usually no spiral thickenings (Figure 2A,B). Although no formal study has been carried out, the venom reservoirs and primary ducts of those ichneumonoids that paralyse their hosts appear to be more heavily chitin-lined and potentially structurally re-inforced. The most likely explanation is that their venoms have basic physiological effects on neuronal or neuromuscular transmission and so would be expected to have adverse effects if they came into contact with the parasistoid’s own neuromuscular system. It would be interesting to know whether such parasitoids have any additional innate immunity to the effects of their particular venoms.

Members of the cyclostome lineage of braconids above the Rhyssalinae, but including some Aphidiinae [66], are also characterised by having Type 1, heavily muscularised venom reservoirs. The VGs may open directly into the reservoir, either apically or medioposteriorly (most Ichneumonidae), or open at the extreme posterior end of the reservoir (Figure 2C,D) or in some braconines, into a small, separate swelling between the reservoir and primary venom duct (Figure 2E). Members of both the Braconinae and Doryctinae (and some other related cyclostome braconids) additionally have many smaller glands opening into the primary venom duct [67,68] but the role of their products is completely unstudied. Another potentially important morphological feature is that in many Rogadinae (koinobiont endoparasitoids but of potentially recent evolutionary origin [69]) have a structure comprising a distally directed comb of cuticular projections at the posterior end of the venom reservoir where it joins the primary oviduct [63]. This has been interpreted as one-way valve system preventing secretions from glands associated with the primary venom duct interacting with primary VG secretions in the venom reservoir.

The above features suggest that the venom in the reservoir may not be the final active product and that the venom primary duct glands probably release enzymes that convert a pre-venom into the final product. Even with a small wasp such as *Habrobracon*, it ought to be practicable to separate the venom apparatus (perhaps under liquid paraffin) into glands, reservoir and primary venom duct, and so obtain separate and combined extracts of each.

The venom glands of two *Psyttalia* species (Braconidae: Opiinae) are shown in Figure 3. The venom apparatus of *P. concolor* (Figure 3c,d) is of particular interest because it possesses a small spherical structure posterior to the reservoir (Rg in Figure 3c,d). This structure has been referred to as a valve [70], but no evidence for that has been found [62]. More recently, it has been proposed to be a gland but without any histological evidence [55]. In specimens macerated in caustic alkali, this basal bulb is pigmented and apparently walled with a thick chitin layer, so a glandular function also seems improbable, and its function remains uncertain. It is not present in any other congeners investigated.

VG histology has been investigated in several wasps used in venomological investigations including: *Asobara* spp. [64], *Habrobracon hebetor* [71], *H. juglandis* [67], *Bracon vulgaris* [72], *Microctonus hyperodae* [73], *Opius caricivorae* [74], *Diadromus collaris* [75], *Pimpla turionellae* [76,77,78], and several species [79]. The VG cells are class 3 in the terminology of [80], i.e., with the gland cell penetrated by a cuticular ductule or canal which runs into and is secreted by a separate ductule or canal cell. Transmission electron microscopic studies consistently reveal extensive rough endoplasmic reticulum, numerous Golgi apparatuses, and vesicles [64]. The actual excretory apparatus has dense microvilli. It should be noted that the structure labelled as Dufour’s gland in the dissection micrograph presented by [72] appears to be a duplicated venom-gland and reservoir, rather than the Dufour’s gland which is a single tubular gland that does not join with the primary venom duct [2,5,65,67].

### Viruses and Virus-Like Particles Produced in Venom Glands

Polydnaviruses are produced in large quantities in the calyx glands, which are swollen distal parts of the lateral oviducts, and not in the VGs (see Figure S7 in [19]). In non-PDV ichneumonoids, a few species produce virus-like particles (VLPs) in their VGs and release these into the host during the oviposition process and one species has a symbiotic virus associated with the venom glands which is a venom constituent that infects the host upon envenomation. Unlike the PDVs, these are not always produced in such large numbers, though they may be (Figure 4) [81].

The only true VG-associated virus involved in host regulation is from of the opiine braconid *Diachasmimorpha longicaudata*, a koinobiont endoparasitoid of fruit flies (Tephritidae) [27]. *D. longicaudata* entomopoxvirus (DlEPV), a poxvirus, replicates in cells of the wasp VG and in haemocytes of the host, the Caribbean fruit fly, *Anastrepha suspensa* (Diptera: Tethritidae) [27,82]. The virus induces host immunosuppression, haemocyte blebbing, and apoptosis in the host [82]. Several genes have been sequenced in DlEVP which confirm it to be a pox virus, and still the only mutualistic true virus found to date in a parasitic wasp [83,84,85]. Of eight sequenced genes, the gamma-glutamyltransferase may have direct venom like actions on the host whereas the other identified genes seem likely to be involved in cell infection and viral replication. DlEPV has been tentatively suggested to be derived from group C dipteran EPVs [84].

Unlike viruses, VLPs do not contain encapsulated RNA or DNA but this is not that easy to prove experimentally [86]. VLPs produced in VG cells are documented from the braconids, *O. caricivorae* and *Psyttalia* (as *Opius*) *concolor* [74,87], and in *Meteorus pulchricornis* [88]. Whether these VLPs play a role in host regulation has not been investigated [86]. Interestingly, another study of *P. concolor* and the congener *P. lounsburyi*, failed to detect any VG-associated VLPs despite targeted effort to find them [55].

VGs of the euphorine braconid *Meteorus pulchricornis* produce copious amounts of a VLP (termed MpVLP) that induce apoptosis in haemocytes of its host, *Mythimna* (= *Pseudaletia*) *separata*, both in vivo and in vitro. Further, host granulocytes exposed to MpVLP show reduced spreading with retraction of filopodia within 15 min and it is thought that this rapid action helps protect the wasps’ eggs from being encapsulated. Plasmatocytes appear to be unaffected [88,89]. Following the creation of a cDNA library, two MpVLP associated components were identified that were involved in inhibiting host haemocyte spreading [52].

The VLPs of *Microctonus aethiopioides* (Euphorinae) (MaVLPs) are produced in the ovarian epithelial cells and not in the venom glands, and are present the ovarian fluid [90]. The congeneric *M. hyperodae* and *M. zealandicus* lack similar particles [91].

Venom of the alysiine braconid, *Asobara japonica*, contains VLPs that cause host *Drosophila* larval death but whose effects are counteracted by other wasp oviposition products (see Section 4.1.1). Such particles are absent from the venoms of the other *Asobara* species investigated.

It seems that with some exceptions, such as DlEPV, that presence of viruses and of VLPs in the venom (and possibly ovarial system) is labile in many parasitoid species and may depend on the strain. VLP production does not seem to be generally characteristic of all members of a given genus.

## 3. Overview of Venom Effects

### 3.1. Paralysing Effects of Venoms

Although both temporary and permanent paralysis effects are very widespread in both families, we actually have rather few detailed observations [92].

#### 3.1.1. Temporary (Transient) Paralysis

A lot of kionobiont endoparasitoids attack their hosts in two stages. The first involves injecting a paralysing venom and usually the second, oviposition into the now immobilised host larva, which might involve very precise placement of the parasitoid’s egg that would have been difficult if not impossible had the host been constantly wriggling. Temporary paralytic venom effects are widespread among both braconids and ichneumonids, probably far more so than the literature suggests because many observers fail to mention it. It has been reported to occur in various members of the braconid subfamilies Agathidinae, Alysiinae, Aphidiinae, Euphorinae (Meteorini), Opiinae, Rhysipolinae and Rogadinae [69,93,94,95,96,97,98,99,100,101,102,103,104] and the ichneumonid subfamilies Adelognathinae, Campopleginae, Pimplinae and Tryphoninae [105,106,107,108,109,110].

In most cases the main purpose of temporary paralysis seems to be to facilitate oviposition which might take a little longer than just a quick envenomating stab with the ovipositor. Part of this may be to prevent a potential host from escaping after the initial attack. For example, several species of blowfly larvae, (e.g., *Sarcophaga* spp.) are rapidly and completely paralyzed by envenomation by the alysiine braconid *Alysia manducator* but recover to resume normal activity after just a minute [93,111]. In other cases, temporary paralysis may also facilitate host-feeding by the wasp [109], and it has also been suggested that transient paralysis caused by endoparasitoid envenomation may help to reduce self-superparasitism rates because immobile hosts are rejected or not noticed or obviously already attacked [101,112].

Even when temporary paralysis is observed you cannot always be sure that it is the result of venom action, for example some parasitoids oviposit directly into the host’s fused thoracic ganglia [95], and it is easy to imagine that the simple physical disruption could lead to paralysis—a bit like the effect of a rabbit punch.

Many members of the rogadine braconid genus *Aleiodes* cause temporary paralysis of their caterpillar hosts, and for a while Mark R. Shaw (pers. comm.) had thought it was probably restricted to those with hosts in low vegetation because they would seem to have a better chance of making it back on to their host plant if the paralysis caused them to lose their grip. However, temporary paralysis is also induced in some tree-feeding hosts.

#### 3.1.2. Long Term/Permanent Paralysis

Better known is that idiobiont ectoparasitoids of mobile larval host stages that are paralysed by the action of the female venom, though some larger species do attack less mobile prepupal and pupal host stages. The paralysis these parasitoids induce is usually considered to be permanent but very few cases have been studied in enough detail to know whether the paralysis might wear off, albeit with the host probably not surviving even if not actually parasitised. These venoms are nearly ubiquitous among all idiobiont ectoparasitoid families. In the Braconidae these include members of the large subfamilies Doryctinae and Braconinae as well as other smaller ones such as Rhyssalinae and Hormiinae s.l. although published observations are few. The potencies of some of the venoms from some species (possibly most) appear to be very high and the action can be very rapid, for example, venom from the doryctine *Spathius agrili*, a parasitoid of the emerald ash borer (*Agrilus planipennis*: Buprestidae) [113,114]. Very high venom potency is probably a pre-requisite in many cases of gregarious ectoparasitoids such as *Spathius agrili* and *Habrobracon hebetor* because the host larvae can be more than 100 × the mass of the adult female wasp. Long term paralysis might also serve to reduce superparasitism rates as location and attack of concealed hosts often relies heavily on vibrations caused by host feeding or other movements [115].

However, the situation is complicated because it is becoming apparent that oral secretions from larval parasitoids might have paralysing actions on the host too, especially if the egg deposition takes place far from the proximity of the host. DLJQ was made aware of this by personal communication from S. B. Vinson (Texas A&M University, College Station, TX, USA) but the experiments appear not to have been published. It seems likely that this is the case also with the recently described *Bracon garugaphagae* which is entomophytophagous on gall-forming psyllids (Hemiptera) [116]. It would be most interesting to know whether the paralysing components in the larval oral secretions are in any way related to the paralysing venom components from the adult females of the same or related species.

### 3.2. Venoms and Nutritional Regulation

One of the most interesting effects of parasitoid wasp envenomation of hosts is that it often induces widespread changes in host biochemical composition (e.g., abundances of in protein, lipid, carbohydrate and other host components), or at least changes in the haemolymph composition. When the braconine, *Bracon mellitor* envenomates the boll weevil, *Anthonomus grandis*, the physical appearance of the host haemolymph changes dramatically after two days. Within 6 min of envenomation there is a rise in haemolymph free amino acid content but this declines after approximately two hours [117]. However, after 4 days there is a second rise in haemolymph free amino acids and a paralleled decline in protein. These changes in protein, amino acid and sugar composition can easily be interpreted as having evolved to make the host more nutritious to the developing parasitoid larva.

Artificial envenomation by *Pimpla turionellae* of prepupal *Galleria mellonella* larvae and pupae causes different effects on the different stages as well as showing differences in the dose-effect relationship [118,119,120]. Briefly, low venom doses lead to increases in both larval and pupal haemolymph carbohydrate, and dose dependent decrease in pupal lipid content. However, the cause of the haemolymph lipid reduction is unknown and could result from a reduction in **“**fatty acid synthesis, rapid accumulation of lipid in fat body, or inhibition of lipoprotein synthesis that are involved in host immune responses**”** [120]. No novel host haemolymph protein bands have been detected in this following envenomation or venom injection so all the effects appear to be quantitative [121].

Venom of *Bracon mellitor* affects both free amino acid and protein titres of a host, the boll weevil, *Anthonomus grandis* [117]. Both immediately after envenomation (6 min) and four days subsequently, free amino acids increased and protein titres decreased. As wasp eggs hatch within 24 h the changes would not be of any immediate benefit to 1st instar larvae. The host haemolymph changes were interpreted as being of nutritional benefit to the *Habrobracon* larva. Although one study [122] of *H. hebetor* envenomation (of larvae of the Indian meal moth, *Plodia interpunctella*) failed to detect any changes in host haemolymph protein composition up to 72 h, a more recent one (on *Ephestia kuehniella* larvae) showed that 12 proteins were down-regulated and five up-regulated, but no detectable (with SDS-PAGE) novel proteins were expressed [123]. It could be that the venom effects vary between host taxa, though amino acid titres have only been investigated in one as yet.

An alternative way of examining the effects of parasitisation or envenomation on the food-resource qualities of the host is to compare protein expression profiles between control and parasitised/envenomated hosts using proteomics, e.g., [124].

#### Host Castration

Host reproductive organs (ovaries and testes, or their imaginal buds) are of no use to the parasitoid larva and, if they are consuming resources, would be better destroyed (castration) [111]. The best studied case involves envenomation by the aphidiine braconid, *Aphidius ervi*. Serine proteases and especially gamma-glutamyl transpeptidases in *A. ervi* venom are injected into the host at oviposition and have a role in the modulation of the aphid physiology including inducing apoptosis of host ovarial germarium and thus arresting host reproduction [46,48]. In addition, all but the most developed embryos show signs of degradation [125]. Host castration might be a common feature of all aphidiines (at least in Aphidiini) as it has also been demonstrated to be caused by *Lysiphlebia japonica* attacking *Aphis gossypii* [124].

Male, but not female *Drosophila melanogaster* larvae have smaller gonadal imaginal buds when parasitised by the alysiine *Asobara citri* [126]. As testis formation in the larva is thought to require more resources than ovary formation it seemed reasonable that this partial castration might have evolved to divert resources and produce large parasitoid offspring. However, no relationship was found between host testis volume and parasitoid dry weight.

Envenomation by the euphorine braconid, *Microctonus hyperodae* which is a parasitoid of adult weevils causes the host to stop laying eggs within a few hours [127]. However, it is not known whether this is caused by venom or by the associated VLPs (MaVLPs) released in the ovarian tissues.

### 3.3. General Effects on Host Immune System

When a parasitoid pierces a host with its ovipositor to inject venom and/or lay an egg, it creates a wound through which the host might get infected. Later in parasitoid development, the host digestive tract will often be compromised and so potentially release bacterial and fungal pathogens into the haemocoel. Both could be harmful to the parasitoid. One might expect therefore, that parasitoid venoms might contain antimicrobial toxins. Any beneficial effect on the host immune system would necessarily have to be targeted against pathogens because up-regulating it generally would not be good for the parasitoid.

When *Pimpla rufipes* venom is injected into tomato moth, *Lacanobia oleracea* (Lepidoptera: Noctuidae), pupae the host shows increased susceptibility to the fungal entomopathogen *Metarhizium anisopliae* (Fungi Imperfecti: Deuteromycotina) as would be expected if the venom has caused a general reduction in host immune competence [128]. Similarly, venom increased susceptibility to two entomopathogenic bacteria, *Bacillus cereus* and *Beauveria bassiana* [129].

## 4. Review of Studies on Ichneumonoid Wasp Venoms

### 4.1. Braconidae

#### 4.1.1. Alysiinae

The Alysiinae and Opiinae (see Section 4.1.7) form a monophyletic clade, all members of which being koinobiont endoparasitoids of Diptera larvae, emerging from the fly puparium which is formed by the cuticle of the final instar maggot [5]. Three species of *Drosophila* parasitoids in particular, have been studied in depth. These are the European *Asobara tabida*, the North African *A. citri* [100] and the East Asian *A. japonicum*. Marked differences between all three species have been observed in their mechanism of avoiding encapsulation and host paralysis [130,131]. The active components of *Asobara* venoms are, at least largely, proteinaceous, and the venom is somewhat acidic. In addition to large proteins, *A. tabida* venom also contains some small peptides [132].

The avoidance of encapsulation by *A. tabida* eggs does not rely on venom but instead on the sticky (fibrous) nature of the egg’s exochorion [133,134,135,136]. Normally a large proportion of the parasitoid’s eggs become embedded within various host tissues including digestive tube, fat body, tracheae [137]. Host haemocyte counts are not affected by parasitism. In contrast, *A. citri* eggs are not sticky and float freely in the *D. melanogaster* host haemocoel but are rarely encapsulated [100,130] but unlike parasitism by *A. tabida*, the host haemolymph count is markedly reduced as a result of a presumed chemical action on the anterior lobes of the haematopoietic organ. The ability of *Drosophila* hosts to encapsulate *A. tabida* eggs is positively correlated with the concentration of host haemocytes [137,138]. Successful encapsulation also depends on the host showing a primary hemocytic response, which gives rise to the amplification of the haemocyte population after immune challenge. In addition, host haemocyte load needs to be large enough that the parasitic egg becomes coated in host haemocytes before it can stick to and become embedded among the host tissues [134,138]. This system has proved useful for study of the evolution of host resistance and parasitoid virulence [139]. The *A. tabida* system has also been investigated from an evolutionary perspective with artificial selection experiments that increase parasitoid virulance and host immunity in separate selection lines [139].

Venoms of *Asobara citri, A. persimilis* and *A. tabida* induce temporary paralysis of the host larva. In the case of *A. persimilis*, the venom immobilises the host within one or two seconds (causing flaccid paralysis) and the host recovers full mobility after approximately 60–90 s [140]. Injection experiments have demonstrated that the transient paralysis is the result of the venom but the strength of the effect varies between parasitoid strains [97].

The most studied and probably most important component of *A. tabida* venom is an aspartylglucosaminidase protein (= N4-(beta-N-acetylglucosaminyl)-L-asparaginase) (AGA) which is its most abundant component and is secreted as a fully functional enzyme [132,141]. This is normally a lysosomal enzyme that is involved in catalysis of N-linked oligosaccharides of glycoproteins during their breakdown and is crucial in the last stage of degradation. They catalyse breaking the bond N-acetylglucosamines from asparagine. Interestingly, a similar AGA is also abundant in the venom of another, but distantly related, *Drosophila* parasitoid, the cynipoid wasp, *Leptopilina heterotoma* (Figitidae) [142]. Using a cDNA library it has been shown that the wasp’s DNA encodes a precursor molecule comprising two different subunits, *α* and *β*, which are assembled into a heterotetramer comprising two of each subunit [141]. The masses of the *α* and *β* subunits are 30 and 18 kDa, respectively, and it is the precursor molecules that are stored in the venom reservoir [132]. Enzymatic activity of the venom AGA was found to be quite sensitive to variations of pH and temperature. Because the venom AGA showed no activity towards entire glycoproteins, its in vivo substrate may be restricted to free glycosylasparagines [141].

The AGAs of *A. tabida* and *L. heterotoma* were sequenced and the structures of the folded proteins estimated [142]. Unlike human AGA only one subunit in each is glycosilated, the *α* subunit in *A. tabida* and *β* subunit in *L. heterotoma*. The two AGAs have similar aspartylglucosaminidase and asparaginase activities and it has been postulated that the latter may be important in affecting host physiology [142]. Human and the two wasp AGAs have a similar domain organisation and share with them key residues for autocatalysis and activity. SDS-PAGE analysis of *A. tabida* venom shows a typical number of protein bands, some of which shift markedly under reducing conditions (Figure 5). Blotting the SDS-PAGE profile on to nitrocellulose enambled immunostaining with two antibodies, one against the α-subunit of human AGA (anti-hAGA) and one against the same subunit (i.e., 30 kDa protein) of *A. tabida* (anti-P30). Both antigens reacted predominantly with the same proteins that were strongly staining in the whole venom PAGE result, and to a lesser extent with a heavier band that represents a heterodimer.

The role played by *Asobara* venom AGA is uncertain, however both are also efficient asparaginases and it has been postulated that the latter may be an important aspect [142]. It has been postulated that AGA causes the production of free aspartate and that this directly affects the host *Drosophila* nervous system as a neurotransmitter [143]. This is based upon the discovery of a selective Na+-dependent excitatory amino acid (aspartate) transporter (dEAAT2) in *Drosophila* nervous tissue that has a much higher affinity for aspartate than for the insect neuromuscular excitatory transmitter L-glutamate [144]. Aspartate is also a potent agonist of glutamate at the insect neuromuscular junction [145], so it is not currently possible to determine which or both actions might be important in the venom’s activity.

*Asobara tabida* venom also induces a small reduction in host haemolymph phenoloxidase activity which presumably reduced the host’s encapsulation ability [146] but this has not been further investigated.

The venom of *A. japonica* in contrast to that of the other investigated *Asobara* species, is lethal against the wasp’s normal hosts, causing permanent paralysis or death [10,131]. The latter case obviously is more complicated because a koinobiont parasitoid cannot survive in a dead host (see below) [10]. Interestingly, *A. japonica* venom is not lethal to non-host *Drosophila* species [10,147]. It does not affect the humoural microbicidal activity of host haemolymph (determined as expression of antimicrobial peptide (AMP) genes) but does alter the behaviour of host haemocytes, inhibiting both spreading and phagocytosis [148]. Further, venom injection leads to a large increase in host serine protease activity in host haemolymph after 4 hrs. This serine protease increase is partially counteracted by the oviduct secretion, and perhaps this is the mechanism for reducing the toxicity of envenomation. The high level of toxicity of crude *A. japonica* venom might also be related to the broad host range of this species compared, for example, to the lower toxicity of *A. rossica* venom. Venom lethality is neutralised when co-injected with lateral oviduct extract (Figure 6) [149].

The situation with *A. japonica* venom appears to be more complicated than in other studied *Asobara* species as it involves VLPs which which seem to be responsible for the lethal nature of the venom on the normal host. The activity of the venom was essentially destroyed by exposure to UV light, by sonication, and by boiling (Figure 6). However, activity was not affected by treatment with trypsin, benzolase (a genetically engineered endonuclease), RNAase or DNAase (Figure 6). That sonication, UV exposure and boiling destroyed the activity suggested that the active agent might be a virus or VLP, and the absence of effect of nuclease enzymes supported the idea that it might be a VLP rather than a true virus.

The active component can be obtained by precipitation of the venom by ultracentrifugation. Centrifugation of the venom showed the active component was absent from a pellet obtained with 120,000 g and was still present in the supernatant. Further centrifugation of the supernatant at 450,000 g recovered much activity in the pellet although most activity remained in the supernatant (Figure 7). Finally, TEM of the active pellet reveals heterogeneous spherical virus-like particles 20–40 nm in diameter (Figure 8). The presence of VLPs in the venom of *A. japonica* and not in any of the other studied species within the same genus seems to indicate that genes for VLP production can be acquired/evolved very rapidly.

#### 4.1.2. Aphidiinae

The entire subfamily are specialist solitary endoparasitoids of aphids (Hemiptera: Aphididae) and several species are important in control of pest aphid species, and more than twenty have been deliberately introduced as biological control agents. Despite their small size and the problems that impose on venom extraction, the venoms and their effects have been studied extensively in the common species *Aphidius ervi* on its host, the pea aphid, *Acyrthosiphon pisum*. Unlike most other insects (at least holometabolous ones) the vast majority of aphids do not encapsulate parasitoids, rather in resistant aphid strains the parasitoid egg slowly disappears as a result of some not understood humoural mechanism [5,150].

Aphidiine venoms induce temporary paralysis in the host aphid, an activity that has been reported in a number of species (two *Binodoxys* species that attack *Aphis glycines* [112]; *Monoctonus paulensis* [151]). In the case of *M. paulensis*, the duration of paralysis varied between host species ranging from two to 23 min [95]. The injection of *A. ervi* venom extract into fourth instar hosts leads to developmental arrest and often host death though boiling the extract greatly reduced its activity, suggesting that the responsible components are proteins [152].

Further, *A. ervi* venom causes destruction of the aphid’s ovaries and presumably this diverts resources so that they are more readily available for the parasitoid larva since it then prevents the development of new aphid embryos [153,154]. The venom also leads to reductions in both embryo number, and the size of the largest embryo, as well as degenerative changes in the embryos themselves [125]. However, it is not known whether these changes are due to the parasitoid’s venom or indirectly the result of starvation or perhaps host hormonal changes.

A chromatographic fraction of *A. ervi* venom enriched with two proteins with masses of 18 and 30 kDa reproduces the host castration effect by inducing apoptosis of cells in the germaria and ovariole sheath of the host aphid [46]. cDNA clones were sequenced and found to match cDNA sequences from a VG library corresponding to a 541 amino acid protein with a prediced mass of 56.9 kDa. This protein contained the amino acid sequences of both the 18 and 30 kDa constituent monomer proteins. The protein showed a significant level of sequence identity with γ-glutamyl transpeptidases and is named Ae-γ-GT. γ-glutamyl transpeptidases are important in the metabolism of glutathione and are normally membrane-bound heterodimeric proteins with a large and small subunit. The subunits derive from post-translational processing of a larger single protein precursor molecule. Unlike typical γ-glutamyl transpeptidases, that in the venom is secreted as a soluble molecule. The detection of γ-glutamyl transpeptidase activity in the active chromatographic fractions corroborates the hypothesis that this is the molecule causing the apoptosis and hence host castration. Venom and ovarian extracts also inhibit host moulting, venom apparently having the larger effect though quantification is not very exact [152]. In normal ovipositions it is thought that *A. ervi* injects approximtely 4 ng of Ae-γ-GT into each host [155].

A proteomic analysis based on a cDNA library from *A. ervi* identified five further putative venom proteins, though γ-glutamyl transpeptidase was represented by the largest number of expressed sequence tags (ESTs) (Table 3) [48].

An enolase has also been found to be present in the ‘oviposition ejecta’ of *A. ervi* using a very nice technique with chitosan beads as surrogate aphid hosts [155]. However, no enolase was detected in VG extracts and it was suggested that the enolase, which is similar to that released in the host by *A. ervi* teratocytes, might be associated with the egg chorion and not the venom and involved in immunosuppression.

#### 4.1.3. Braconinae

This is a huge cosmopolitan subfamily whose members are mostly idiobiont ectoparasitoids of holometabolous insect larvae. The venom of only one species, *Habrobracon hebetor*, has been investigated in any detail, with a few studies also on *H. brevicornis*, *H. gelechiae* [156], *H. nigricans* [51,157] and *Bracon mellitor* [117]. As regards work published as having been carried out on *H. hebetor*, it should be noted that there has been some taxonomic confusion so it is possible that some laboratories were actually working on the related *H. brevicornis* that was for a long period treated as a junior synonym of *hebetor* [158]. There have also been a few studies on *H. brevicornis*, and the two species appear to have different host preferences [159].

*Habrobracon hebetor* was once a common laboratory animal and indeed was sent into space on a space mission [160]. It is not surprising then, that its paralysing venom soon became a topic of investigation, in some cases with the idea that it might lead to the development of novel insecticides [161,162,163,164,165,166], as well as tools for the neurophysiologists [167]. It is easy to culture on various hosts [168], notably stored product pyralid moth larvae including the Indian meal moth (*Plodia interpunctella*) and Mediterranean flour moth (*Ephestia* (=*Anagasta*) *kuehniella*) and also on the greater wax moth (*Galleria mellonella*). However, the venom has been tested against a wide range of other Lepidoptera (Table 4) as well as various other insects [169,170,171]. There is variation in its effectiveness between hosts as it is effective against many [172] and it is lethal to some [173]. At sublethal doses VG extract reduces the growth rate of gypsy moth (*Lymantria dispar*) larvae [174]. However, the venom seems to be specific to Lepidoptera and only has a a paralytic effect on members of the Diptera, Hymenoptera and Hemiptera if administered in large doses [175]. *Habrobracon hebetor* will also attack hosts that have previously been parasitised by some endoparasitoids, for example, when it envenomates *E*. *kuehniella* containing larvae of the campoplegine *Venturia* (as *Nemeritis*) *canescens* it causes complete developmental arrest of the latter, but not paralysis, the effect being more rapid on earlier *V. canescens* larvae [176].

The venom of *H. hebetor* and an “Indian *Bracon*”, which seems likely to have been *H. brevicornis*, were shown to be still active following desiccation and rehydration, and that of the “Indian *Bracon*” was still active in females that had been dead for six months [177]. The venoms were rapidly inactivated by treatment with a proteolytic enzyme mixture (pancreatin) which suggested that the toxins were proteinaceous. The venom starts to be secreted approximately two-thirds of the way through the pupal stage so freshly emerged females of the Indian wasps had a full venom compliment.

*Habrobracon hebetor* envenomation causes flaccid paralysis in a wide range of caterpillar species, and leads to the death of many cases. The venom is exceedingly potent against a range of hosts. It has been estimated that one part *Habrobracon* venom to 200 million parts host hemolymph of *G*. *mellonella* is enough to cause complete paralysis [161,166]. The time taken for complete flaccid paralysis to develop varies significantly between host species [172]. A typical envenomation dose also causes a reduction in respiration of approximately 50% [178].

The first work on its paralysing venom was by Beard who observed that whilst the caterpillar host was paralysed, contractions of the heart and gut muscles were not affected [161]. Host heart contractions are now known to be purely myogenic [179,180]. This pointed to a possible neuropharmacological specialisation though at that time the peripheral excitatory neurotransmitter in insects was not known.

The neuromuscular activity of *H. hebetor* venom has subsequently been investigated by several groups [181,182,183]. Using whole and ligated caterpillars of the giant silk moth *Philosamia cynthia* (Saturniidae) as a conveniently large, factitious host, it was shown that injection of *Habrobracon* venom prevented electrically induced and spontaneous contraction of peripheral muscles [162]. The venom was only effective in the ligated body region that it was injected into indicating that it blocked peripheral excitatory neuromuscular transmission and had no effect on central nerve transmission. That after paralysis had been induced, much stronger electrical stimulation could still evoke muscle contraction showed that the muscle itself was not affected. Investigation of the effect of the venom on *P. cynthia* neuromuscular preparations revealed that venom leads to a gradual reduction and finally, elimination, of miniature excitatory post synaptic (glutamate) potentials (Figure 9) [184]. In mealmoth larva (*Ephestia kuehniella*) and of the locust (*Locusta migratoria*) preparations, the venom was shown to have no effects on either neuronal conduction, muscle resting potential nor on GABA-mediated inhibitory neuromuscular transmission in insects. Most importantly the post-junctional sensitivity of the muscle fibres to the excitatory neurotransmitter L-glutamate was not affected showing that the venom’s action was presynaptic. In locust and *Ephestia* preparations, the spontaneous release of individual presynaptic glutamate vesicles that cause miniature excitatory postsynaptic junction potentials was reduced by between 50 and 99% [185]. The block of neurotransmitter exocytosis at the neuromuscular synapse does not depend on the influx of Ca^2+^ ions [186].

Thus, the data indicate that the main process involved in the venom-induced paralysis is the block of exocytosis of presynaptic neurotransmitter (L-glutamate) vesicles [162,181,185,186].

A putative endonuclease (34 kDa) and two putative phospholipases (phospholipase A2, with molecular weights 17 and 21 kDa) have been patented with a claim to use of these for insect pest control [165].

Some central nervous system involvement in the action of *Habrobracon* venom cannot be ruled out altogether because it has been shown to cause a 40% reduction in 2-cholinesterase activity in the larvae of the rice moth, *Corcyra cephalonica* [187].

The following gives a brief summary of the arduous and rather confusing path to studying the main paralytic proteins. A fraction with paralysing activity thought to be a single protein of approximate mass 61–62.7 kDa and an isoelectric point of pH 6.8. was purified from *H. hebetor* venom by a combination of ion exchange chromatography on DEAE-Sephadex A-50, gel chromatography on Sephadex G-100, electrophoresis on polyacrylamide gel and gel chromatography on Sephadex G-75. The protein was however very labile and although they achieved a 28-fold level of purification they only recovered about 2.5% of the original biological activity [188]. At about the same time, using using a bulk extract from 10 g of female *H. hebetor*, and somewhat similar extraction/purification process, resulted in two Sephadex A-50 fractions with paralysing activity, called fractions A and B, corresponding to proteins of approximate masses of 42 and 57 kDa, respectively. These were later named A-MTX, and B-MTX [189]. Again, the extracts showed only a tiny fraction of the original activity. Firstly, given the apparent differences in mass it might at first be supposed that they are different proteins to the one extracted by Visser et al. [189]. However, the heavier B protein’s mass is comparable and its lightly lower mass might be attributable to impurities in the samples [183]. Venom proteins A and B had produced significantly different dose response plots and showed differential activities against a number test insect species.

After many attempts by various groups, five high molecular weigh proteins (circa 73 kDa) have been isolated from *H. hebetor* venom, named Brh-I through Brh-V, though Brh-IV apparently comprised two separate toxins [170]. All these proteins were found to be quite labile under various extraction procedures. Two of the isolated proteins (Brh-I and Brh-V) showed very high insecticidal activity, the first being the most toxic with an LD_50_ < 2 pmol/g against *Heliothis virescens* and 0.002 μg/g against *G. mellonella*. For Brhs I, III and V the authors also managed to sequence between 22 and 41 consecutive amino terminal amino acids. Their extraction procedure utilised anion exchange chromatography and was apparently more successful than previous studies. Further, the 73 kDa toxins were shown to be responsible for the greater part of the activity of whole venom, but there action was slow with it taking approximately 1 h to induce flaccid paralysis rather than a minute or so [106,161,190,191]. Thus it seems reasonable to assume that the rapid onset of paralysis in naturally envenomated host caterpillars is due to some of the lower molecular mass toxins. A relatively low molecular weight (c. 18 kDa) neuro-active component has recently been isolated [192].

Envenomation of *G. mellonella* by *H. hebetor* leads to both a reduction in host haemolymph phenoloxidase activity and a reduced number of haemocytes showing phenoloxidase activity [193]. Envenomation also leads to reduced haemolymph reactive oxygen species as measured by CP-H (1-hydroxy-3-carboxy-pyrrolidine) spin trap, and halved the rate of melanisation of white plastic implants (2 mm long and 0.5 mm diameter) that were injected into the host haemocoel and retrieved after two hours (Figure 10) [194]. The latter effect is potentially also associated with reductions in total and changes in differential host haemocyte counts [195]. Envenomation of *P. interpunctella* by *H. hebetor* results in an increase in host hemolymph phenoloxidase activity which is thought to increase the host’s resistance to microbial infection. However, once the parasitoid larvae start feeding there is a decrease in host hemolymph PO activity, suggesting that parasitoid larval salivary secretions may be entering the host and suppressing its immune system and thus interfering with melanisation at the feeding wound sites [196].

A venom gland transcriptome from *H. hebetor* yielded full length open reading frames for calreticulin, an acid phosphatase type protein (similar to Acph-1) and an arginine kinase [49]. The deduced amino acid sequences of the calreticulin and acid phosphatase genes matched closest to sequences from two opiine braconids for which relevant draft genomes were available, *Diachasma alloeum* and *Fopius arisanus* [197,198]. It was postulated [49] that the arginine kinase may be involved in host paralysis, since the same enzyme in a spider wasp (Pompilidae) has a paralytic action on its prey [199]. Acid phosphatases are present in honeybee venoms [200] and in a wide variety of organisms though despite much research their function is still largely unknown [201]. A whole wasp transcriptome was compared to the results from [49] allowing cloning and sequencing of three target venom genes: venom acid phosphatase Acph-1 like protein, tryptase-2 and CTD nuclear envelope phosphatase proteins [50].

Venom extracted from reservoirs has been shown to be toxic to insect cell lines [28]. Cells treated with venom changed shape (“rounded up”), swelled and died. Three different cell lines were investigated: from the moth, *Spodoptera frugiperda*, the beetle, *Tribolium castaneum* and the mosquito *Aedes aegypti*. Interestingly the LC_50_ and especially the LC_99_ for the lepidopteran cell line were markedly lower than for each of the others. The time taken to kill 50% of target cells was also markedly faster for the *S. frugiperda* cell line, by factors of 15 and 22. These results suggest some specificity for cells from a real potential host insect compared to insects that have no potential host parasitoid relationship with the wasp. It is not yet known whether the active components against the cell lines are the same as any of the toxins described above. One possibility indicated by the swelling of the target cells is that the component involved causes cell membrane porosity.

Recently, proteotranscriptomics has been applied to the venom of another closely related braconine, *Habrobracon nigricans* (as *Bracon*), which among other hosts can be cultured on *Spodoptera littoralis* [51]. Venom proteins were separated on SDS-PAGE gels (Figure 11), and protein bands excised and identified using LC/MS-MS. A summary of the identified and putatively identified venom proteins is shown in Table 5. Only a few of the venom transcripts were found to be shared with *H. hebetor*, these being phospholipases (involved in host paralysis and patented by [165]), arginine kinase and venom acid phosphatase.

Then RNA was extracted from venom glands, female wasps with venom glands removed, and male wasps. The extracts were analysed using qRT-PCR to quantify the expression of particular venom proteins [Becchimanzi et al. 2020]. The results for selected components are shown in Figure 12.

#### 4.1.4. Euphorinae—Meteorini

Meteorini (sometimes treated as a full subfamily) are koinobiont endoparasitoids mainly of Lepidoptera caterpillars with a few species attacking (mostly) concealed Coleoptera larvae; only the former have been investigated.

*Meteorus pulchricornis* has one of the broadest host ranges yet described amongst koinobiont braconids, attacking hosts in as many as 12 families of Lepidoptera [202], and it also has an unusually large venom reservoir [203], though whether these two things are related is unknown. The venom induces host haemocyte apoptosis [88] and inhibits haemocyte spreading in *Mythymna separata* (as *Pseudaletia*) (Lep.: Noctuidae) [89].

Virus-like particles (MpVLPs) are abundant in the lumens of the VG filaments and in the venom reservoir. These particles are composed of single-membraned vesicles of about 150 nm half or completely filled by electron-dense material but do not appear to contain DNA or RNA [88]. The injection of purified MpVLPs into *Mythimna* (=*Pseudaletia*) *separata* caterpillars reduced the ability of host haemocytes to encapsulate fluorescent latex beads by approximately 90%, and induces haemocyte apoptosis [88,204]. MpVLPs have also been shown to affect *Cotesia* (gregarious microgastrine braconid parasitoids) development and survival in cases of multiparasitism and of experimental injection of wasp products [205]. MpVLP injection leads to significant (*p* < 0.05) increases in host pupation rate and reductions in the numbers of *Cotesia kariyai* cocoons that emerge, whereas venom alone had no significant effect (Figure 13).

Arbitrary sequencing of clones based on a conventional cDNA library construction from the VG filaments, which produce both venom and the MpVLPs, yielded 473 independent cDNA clones which formed 228 clusters [52]. BLAST2GO (BLAST to gene ontology) analysis of the clusters allowed 105 of the sequences to be annotated. Twenty of the clusters were fully sequenced (summarised in Table 6). Twelve of these were highly adult-specific and two were associated the the MpVLPs. RNA interference experiments were carried out to produce MpVLPs that were deficient in these two factors. In in vitro assays of the effects of the VLPs on haemocyte spreading, the modified VLPs were found to be less effective at inhibiting haemocyte spreading than un-modified ones.

#### 4.1.5. Euphorinae—Perilitini

Venoms have been investigated in two species of *Microctonus*, *M. aethiopoides* and *M. hyperodae* which are both important biological control agents that have been introduced into New Zealand in attempts to control the lucerne pest weevil, *Sitona discoideus*, and the Argentine stem weevil, *Listronotus bonariensis*, respectively. Euphorines other than Meteorini have an unusual biology in that they are parasitoids of adult holometabolous insects and nymphs and adults of paurometabolous ones.

Sequenced and assembled RNA transcripts from dissected VGs of *M. hyperodae* using cDNA cloning, and from *M. aethiopoides* (Morocco biotype) using direct cDNA sequencing, have been investigated [53]. *M. hyperodae* yielded 374 contigs that were identifiable and 203 with BLAST search matches with *p* < 1 × 10^−5^ [53]. Of these, eight were chosen for full length sequencing and included sequences matching neutral endopeptidase, lysosomal thiol reductase, calreticulin and a heat shock protein. *M. aethiopoides* yielded 757 contigs with BLAST search matches with *p* < 1 × 10^−5^, and the most abundant transcripts are summarised in Table 7.

#### 4.1.6. Macrocentrinae

Macrocentrine braconids are polyembryonic, solitary or gregarious, koinobiont endoparasitoids of Lepidoptera caterpillars. Only one species, *Macrocentrus cingulum*, has been studied from a venomological point of view. *M. cingulum* is a parasitoid of the Asian corn-borer moth *Ostrinia furnacalis*, a major pest of maize (*Zea mays*) and various other crops in Asia and Africa. It has been known for some while that *M. cingulum* eggs avoid encapsulation passively, i.e., without the need for venom, as a result of their fibrous outer layer [206]. The wasp preferentially attacks younger instar larvae that have lower haemocyte titres; encapsulation of the egg is more likely to occur in older hosts, but at any given host stage there is no difference in haemocyte count between parasitised and unparasitised larvae [206]. After hatching, the extra-embryonic membrane is vital for avoiding encapsulation [207].

The venom of *M. cingulum* has been analysed using SDS-PAGE chromatography and found to contain three proteins in substantial amounts, with masses 45 kDa, 64 kDa and 97 kDa [208]. Hemomucins play crucial roles in the passive avoidance of the host immune response system [54]. Correspondingly, *M. cingulum* has fewer venom protein genes than many other hymenopterans, especially esterases (Figure 14) [54].

Whereas *M. cingulum* venom appears to have little effect on host haemocyte count in vivo or on haemocyte spreading and viability in vitro, ovarian proteins do cause a significant suppression of encapsulation of Sephadex A-25 beads [209]. It has been postulated that the venom might cause temporary host paralysis [209] but the authors state “... although this remains to be elucidated in further studies”. We cannot find any further information on this.

Expression of a prophenoloxidase transcript in plasma, haemocytes, fat body and midgut of *O. furnacalis* has been found four hours post parasitisation [210]. However, it was previously demonstrated by the injection of calyx fluid and venom, both separately and combined, that the prophenoloxidase effect results from the calyx fluid and not the venom [211].

The related braconid, *Charmon extensor* (Charmontinae, formerly in Homolobinae) is a Palaearctic koinobiont Lepidoptera larval endoparasitoid with an abnormally large host range including members of the Gelechiidae, Tortricidae, Oecophoridae, Pyralidae and Erebidae [212]. It would be interesting to know whether it employs a similar defense strategy to *Macrocentrus*.

#### 4.1.7. Opiinae

All Opiinae are koinobiont endoparasitoids of Diptera maggots. Many species of Opiinae have been utilised successfully for the biological control of fruitflies (Tephritidae), notably various species of *Biosteres*, *Diachasma*, *Diachasmimorpha*, *Doryctobracon* and *Psyttalia* [59]. Envenomation by both *Diachasmimorpha longicaudata* and *D. tryoni* rapidly causes complete temporary paralysis, the effect lasting for approximately five or two minutes, respectively [98]. The most notable aspect of these wasps from a host-parasitoid interaction point of view is that they seem to house a several symbiotic viruses, at least some of which are probably injected into hosts as part of the wasp’s venom, and likely benefit the wasp by infecting host cells [82,87,213,214].

Surprisingly, despite their economic importance, not a single study of their venom chemistry has been published to date. Interestingly though, several papers have reported on the presence of viruses and VLPs in VGs of opiines [74,87,213]. It is not known whether these play a role in overcoming host responses to parasitism [86].

As well as VLPs, *Diachasmimorpha longicaudata* venom glands contain rod-shaped viral nucleocapsids which are passed to the host during oviposition [215]. The nucleocapsids enter and proliferate within haemocytes of the host *Bactrocera dorsalis*. *D. longicaudata* VGs also host a mutualistic entomopox virus (referred to as DlEPV) [27,213,216]. DlEPV is transmitted to the host during oviposition and replicates in both wasp and host tissues, and has adverse effects on host haemocytes [82]. Whilst DIEPV is not essential for successful parasitism [81], it is highly virulent to the host, and wasps lacking the virus are far less successful at parasitism than wasps with DlEPV [81]. Many but not all strains of *D. longicaudata* also have a venom gland associated rhabdovirus (referred to as DlRhV) [214]. The DlRhV genome encodes parasitism-specific protein 24 (PSP24), that has been postulated to promote successful parasitism but the potential role of the virus in the host-parasitoid relationship is unknown [214]. Both DlEPV and DlRhV are vertically transmitted in the wasp.

Venoms of two species of *Psyttalia* (*P. lounsburyi* and *P. concolor*) have been investigated and compared using transcriptomic and proteomic methods [55]. Amongst other flies, both species attack the olive fruit fly, *Bactrocera oleae*. SDS-PAGE separation of the venom proteins of two strains of *P. lounsburyi* and one of *P. concolor* show good intraspecific concordance but marked interspecific differences (Figure 15, Table 8). Although not all high ranked proteins occurred in both species, there is a highly significant correlation between the ranks of the 15 shared proteins (rho = 0.716, Spearman’s S = 159.3, *p* = 0.0027), suggesting some commonality in the importance of different processes. In both species, GH1 β-glucosidase is a relatively abundant venom component that has not yet been detected in other parasitoid wasp venoms. It has been suggested that it may be involved in breaking down olive glycosides that may have been sequestered by the host maggot, thus releasing toxic compounds and potentially debilitating the host [55].

#### 4.1.8. Rogadinae

Rogadines s.s. are koinobiont endoparasitoids exclusively of Lepidoptera larvae. Although no chemical or physiological studies have been carried out on the venoms of rogadines, one observation is worthy of mention. In the UK, the wasp *Clinocentrus gracilis* attacks the penultimate and final instar caterpillars of the choreutids, *Anthophila fabriciana* and *Choreutis myllerana* [217]. The initial attack involves a brief sting (c. 0.5 s) which causes complete temporary paralysis within 2 s, full recovery being reached within 10 min. Then the wasp makes a second oviposition attack so it is easy to obtain hosts that have been envenomated but not oviposited in. However, caterpillars that were envenomated never (unless they were very close to ecdysis when stung) progress to the next instar. This was termed ‘delayed inhibition of host development’ and it seems most likely that the *Clinocentrus* venom causes disruption of the hosts ecdysone/juvenile hormone system.

### 4.2. Review of Studies on Ichneumonid Wasp Venoms

In comparison with the Braconidae, there have been very few higher level ichneumonid taxa studied for venom chemistry alone, nearly all detailed work concentrating on the genus *Pimpla* in the subfamily Pimplinae. In contrast, a lot of work has been carried out on various genera of the polydnavirus-containing Campopleginae, but mostly these focus on the interactions of wasp venoms with the polydnaviruses [16].

#### 4.2.1. Ichneumoninae

Members of this subfamily are mostly idiobiont endoparasitoids of Lepidoptera pupae or pre-pupae (just as in the case of the Pimplini, see below). The only species studied to date is *Diadromus collaris*, a parasitoid of the diamond back moth, of *Plutella xylostella*. Parasitised pupae show an increase in the numbers of haemocytes, both plasmatocytes and granulocytes, compared with unparasitised pupae [218]. However, within a day of parasitisation, plasmatocyte spreading was suppressed but granulocytes were unaffected. Isolated crude venom was found to cause disintegration and death of both plasmatocytes and granulocytes [56]. Major changes in fat body cells and general disintegration have also been observed in parasitised *P. xylostella* 72 hrs after parasitisation, but it is not known whether these changes are caused by venom or by larval secretions [219].

SDS-PAGE analysis of *D. collaris* venom showed nine strong protein bands with molecular weights ranging from 9.0 to 52 kDa [56]. Protein bands with masses 50.2, 30.5, 28.2, 25.1 and 12.6 kDa were similar proteins that inhibit host development and immunity in other parasitoid wasp species. Comparison of the transcriptomes of VGs and wasps without VGs showed 4397 genes upregulated in the VGs and 18,147 down regulated (a similar proportion to the microgastrine *Cotesia vestalis*). Because of the similarity between the biology of *D. collaris* and the pimple ichneumonid, *Pimpla rufipes* (see Section 4.2.2), Zhao et al. searched for similarities between the putatively secreted proteins of the former and the venom proteins of the latter [56]. Six matches were found: trehalase, metalloprotease, cysteine-rich venom protein 6, cysteine-rich venom protein 2, and two other proteins. Unfortunately the roles of these are not known but the two cysteine-rich proteins are probably protease inhibitors.

*Diadromus collaris* also hosts an ascovirus which is transmitted to the host, however, the virions appear to originate from the oviduct and not from the venom glands [220].

#### 4.2.2. Pimplinae

This is the most well studied subfamily in terms of venom chemistry and action. The studied species all belong to the Pimplini and are idiobiont endoparasitoids of prepupae and/or pupae of various Lepidoptera. By far the best investigated species is *Pimpla rufipes* (referred to in most of the venom literature by its older synonym *P. hypochondriaca*) but there are also some data for *Pimpla turionellae* (see below). Being idiobiont endoparasitoids, there is no need for the wasp to produce paralysing venoms. Further, egg and pupal holometabolous insect stages are known to generally have weak immune capability—in the case of eggs, there are no haemocytes.

*Pimpla rufipes* venom has a strong effect on host (*Lacanobia oleracea*) haemocytes and even at very low concentrations (1.6 ng/μL) inhibited plasmatocyte spreading, and at 3.2 ng/μL, reduced the capacity of haemocytes to phagocytose *Escherichia coli* by 85% [221]. The exposure of a monolayer of cultured cells (BTI-TN-5B1-4; an insect cell line originating from *Trichoplusia ni*) showed that *P. rufipes* venom caused retraction of cytoplasmic extensions, rounding, swelling and ultimately cell lysis [222]. A small venom protein with molecular weight 13 kDa has been shown to be cytotoxic to sf21 cells, a continuous line developed from the ovary of *Spodoptera frugiperda* (Lepidoptera: Noctuidae) [223]. These effects of venom on BTI-TN-5B1-4 cells were accompanied after 15 min by increases in cellular Ca^+2^ (detected using the sensitive probe fluo-4 AM), and also by a loss of mitochondrial resting potential in as short a time as 5 min post treatment [222]. The role of Ca^+2^ in cell changes and death was demonstrated by adding anti-calreticulin antibodies to the venom, which eliminated the increase in intracellular Ca^2+^, reduced swelling and reduced the number of cells dying. The genes for two of the proteins involved in haemocyte inactivation (vpr1 and vpr3) have been cloned [224] and expressed as recombinant proteins using a standard *Escherichia coli* system [225]. Injection of the purified recombinant product (rVPr1) into cabbage moth (Noctuidae: *Mamestra brassicae*) caterpillars significantly increased their susceptibility to both topically applied and injected spores of the entomopathogenic fungus, *Beauveria bassiana* [226].

The venom of *Pimpla rufipes* is acidic and contains peptides and proteins ranging in molecular weight from approximately 5 to 100 kDa (Table 9) [223,227]. The venom shows proteolytic activity and, based on a cDNA library, includes endopeptidases and aminopeptidases [228]. Interestingly, the isolated venom protein called pimplin has a paralytic action on the host and also on a wide range of insects [227]. Pimplin is a small, heterodimeric polypeptide (143 amino acids, 22 kDa) with a proline scaffold and no similarities with any other known protein. Its component peptides have masses of 10.5 and 6.3 kDa that are linked via a disulfide bond [227,229]. Other identified components included acid phosphatase, phenoloxidases, a laccase, a putative metalloprotease, and various cystein-rich proteins [143,230]. The role of acid phosphatases in venoms is not well understood. They are normally lysosomal enzymes involved in cell degradation and death, but it has been postulated that in snake venoms it may be related to the release of toxic purines [231]. A particular issue with their role in parasitoid venoms concerns the pH. They are optimally active under acid conditions (pH c. 5) but host hemolymph is typically near neutral or slightly alkaline, conditions in which they would be expected to show little activity [229].

Laccases are found in many organisms (plants, fungi, insects). They are multicopper oxidases often with a broad spectrum of activity and catalyze the oxidation of numerous aromatic substrates [238,239]. Their precise roles are still rather uncertain but in insect cuticle they are hypothesised to be involved with cuticle tanning and sclerotization. Laccase-2 is hypothesised to play an important role in insect cuticle sclerotization by oxidizing catechols to quinones, which then catalyze protein cross-linking [240,241]. Laccase-1 is often found in the body fat, mid-gut and Malpighian tubules and may have detoxification functions.

Venom from the related *Pimpla turionellae* has been investigated in a number of papers [57,78,242,243,244,245,246,247]. It is an idiobiont endoparasitoid either of the late instar host larva or its pupa, the latter being preferred [243] Both the venom and uterus gland secretions had negative effects on the haemocytes of *Galleria mellonella*, its natural host [242] and on cultured BTI-TN-5B1-4 cells from *T. ni* (see above) [248]. The secretory products specifically affected plasmatocytes and adipohaemocytes. Both secretions negatively affected haemocytes in large white butterfly, *Pieris brassicae*, pupae, but in that case involving both plasmatocytes and granulocytes. The effects were lesser on other Lepidoptera pupae and no effect of anything was found with larvae of the sawfly *Gilpinia hercynia*, leading the author to postulate that the effects were probably Lepidoptera specific. Envenomation of *G. mellonella* results in a significant reduction in host encapsulation ability as measured by the host’s response to injected Sephadex 50 beads (Figure 16) [246]. At any dose less than the LD_99_ injection of venom into larval or pupal hosts results in a marked decline in the number of circulating haemocytes [247,249] and causing apoptosis and ultimately killing them (Figure 17) [250]. Treatment of isolated host haemocytes to the venom even at very low concentrations (0.001 VRE/μL) induced some vacuole formation in both plasmatocytes and granular cells, however, when exposed for longer than 15 min granular cells were found to be much more susceptible showing greater vacuolisation and death [246].

Venom analyses have shown its composition to be rather different from that of *P. rufipes* [251]. *P. turionellae* venom also causes host paralysis, especially pupal hosts, but it lacks the paralytic peptide pimplin (now called pimplin1) so therefore the action must be caused by other compounds. It contains various proteins and peptides including phospholipase B, and a range of biogenic amines, notably noradrenalin and serotonin; the amides seem to be reasonable candidates for the paralytic activity [229], however, the noradrenalin and serotonin content declines with the age of the adult wasp [251]. Three peptides have been named ‘pimpin2’, ‘pimplin3’ and ‘pimplin4’ (Table 10) [57] and one of these is potentially homologous to Vpr1 from *P. rufipes*.

A proteo-transcriptomic study based on *P. turionella* VG revealed four classes of protein that were potentially involved in encapsulation/melanisation processes, and eight families that had previously been described from *P. rufipes* (Figure 18) [57].

The venom in a single reservoir is estimated to contain 0.04 μg of protein with molecular weights ranging from 20 to 106 kDa [245]. Envenomation causes complete paralysis as in the case of *P. rufipes*, especially in pupal hosts [243]. It also contains various peptides and amines. Two neurotoxic peptides known from honey bee venom: apamin and melittin have been reported [245]. The identifications of both of these was based only on SDS-PAGE chromatography and then reverse-phase HPLC with pure apamin and melittin used as standards, so it is possible that other co-eluting peptides could be involved. Apamin is an 18 amino acid globular peptide which constitutes 2–3% of dry honey bee venom, and selectively blocks SK channels, a type of Ca^2+^-activated K^+^ channel and neurotransmitter induced increases in potassium permeability [252]. Mellitin is a linear, mostly hydrophobic, 26 amino-acid polypeptide and the major pain-causing agent in bee venom. It has a wide range of actions including membrane pore-formation, including blocking ion transport pumps such as Na^+^-K^+^-ATPase.

In various wasp systems, venom serum protease homologues have been demonstrated to impair the phenoloxidase cascade and therefore impair melanisation [253,254,255,256].

*Pimpla turionellae* venom has recently been investigated as a potential source of anti-cancer agents by examining its effects on cell lines derived from carcinomas and time- and dose-dependent cytotoxicity has been demonstrated against brain cancer, glioblastoma cells in vitro [29].

#### 4.2.3. Rhyssinae

Only preliminary, currently unpublished but publicly available combined transcriptomics and proteomics data are available for these large idiobiont ectoparasitoids of woodwasps [58]. Two species of *Megarhyssa* (*M. greenei* and *M. macrurus*) were compared. SDS-PAGE separation of proteins revealed considerably different profiles. Venom gland and reservoir components were used for proteomics and the last three metasomal segments used for RNA extraction. Out of 64 putative venom proteins, 24 were predicted to have potential venom-related functions. The protein profiles obtained were quite different to those of other known parasitic wasp venoms. Proteins, putatively corresponding to ones either belonging to known protein families or with known functions, included: ADAM metalloprotease, chitin-binding proteins, Cu-Zn superoxide dismutase, cyclophilin, dipeptidyl-peptidase IV, lycosyl hydrolase, heat shock proteins, histidine acid phosphatase, leucine-rich repeat protein, lipase, phospholipase A2, protein disulphide isomerase and trypsin-like serine proteases. The study also detected a cellulase and a laccase, both of which in other systems can be involved in the breakdown of cellulose and lignin in woody plant tissues, and at least the former would seem to be an unlikely component of a paralysing venom, though laccase is present in the venom of the pupal endoparasitoid *Pimpla rufipes* (see Section 4.2.2) and must clearly have a different function in that system. Nénon et al. [257] provided some evidence that *M. atrata* may release wood-degrading enzymes during the ‘drilling’ for hosts, though they suggested that these were probably secreted by pores along the length of the ovipositor. Although the micrographs presented look convincing this also seems unlikely since no glands have been reported within the lumens of the ovipositor valves and certainly the lumens of the valves are contiguous with the wasp haemocoel and not with the lumens of the primary venom duct or of the oviduct. A more reasonable hypothesis might be that they are exuded from the ovipositor and originate in either the venom system or from the oviduct system which includes a large uterus (=vaginal) gland at the anterior end of the common oviduct [2]. The possibility of contamination might also need to be considered.

## 5. Suggestions for Further Research Opportunities

As already outlined in the section ‘Difficulties working on parasitoid wasp venoms’, there are many hurdles to overcome in studying the chemistry and physiology at the third trophic level. The following suggestions (Table 11) are a number of as yet unstudied systems which might be rewarding and not too difficult to investigate. The criterion we use here is that the host must be amenable to continuous culture or that the effort required might be outweighed by the potential interest of their venoms.

## 6. Conclusions

What is most apparent is that only a microscopically small proportion of ichneumonoid venoms have been studied in comparison with the vast number of species. Whilst this is not surprising given their diversity, the studies on different wasp taxa have largely been carried out by different research groups with different primary objectives. The taxonomic coverage and biological coverage is so small that it is almost impossible to draw many general conclusions. The available information indicates that venom composition is under strong selective pressures to match particular host-parasitoid associations, and major differences have been found between members of the same parasitoid genera or closely-related genera. In a few cases, one or a few major venom constituents have identifiable effects, but the functions of a plethora of less abundant constituents remains largely unknown or a matter of conjecture based on known functions of similar proteins in other systems [55]. Nevertheless, Table 5 and Table 8 show that a small number of identifiable venom protein types occur in multiple, phylogenetically distantly related species, and it seems likely that they play similar roles in each host.

It is hoped that some of the suggestions for research on additional taxa given in Table 11 will encourage physiological/venom research groups to explore new territory. But even without the study of many more species, a major desire would be to have the same systems investigated across those taxa for which we already have some data. With the exception of a large series of investigations on the *Asobara*/*Drosophila* system, we currently have limited information on whether similar physiological or biochemical strategies are employed different species within the same subfamily or between members of closely related subfamilies. This is where a proteotranscriptomic approach is expected to play a major role. It is potentially applicable to many groups of parasitoids, including ones that will be hard to culture under laboratory conditions, but which might be readily obtained from field collections at an appropriate time of year. For this to be successful it will be essential for physiologists/geneticists to team up with entomologists with taxonomic expertise as well as field craft.

Another major hurdle is that physiological investigations are usually limited by the expertise and interests of individual research groups. For example, the permanently paralysing venom of *Habrobracon hebetor* has been quite extensively investigated even though the precise mode of action is still uncertain. However, the permanently paralysing venoms of *Pimpla* species, especially pimplin from *P. rufipes*, has received no similar investigation. Nor have any of those venoms that induce temporary host paralysis.

## Figures and Tables

**Figure 1 biology-10-00050-f001:**
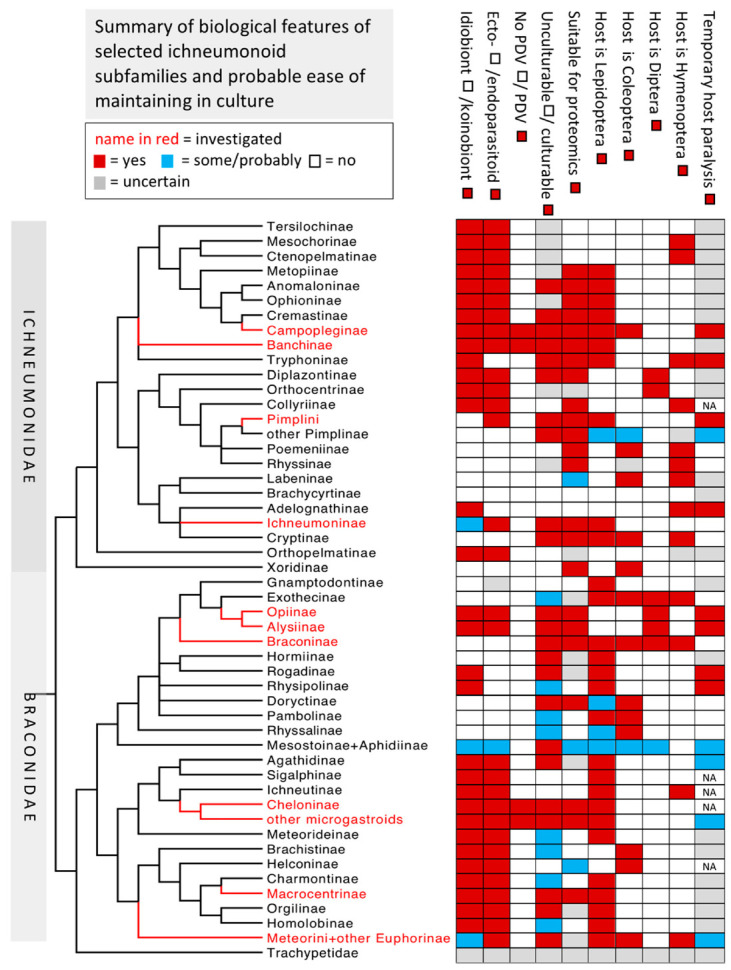
Summary of biological features of selected ichneumonoid subfamilies and probable ease of maintaining in culture. For ease of viewing the following rare subfamilies have been omitted: the braconids—Acampsohelconinae, Amicrocentrinae, Apozyginae, Maxfischeriinae, Microtypinae, Telengaiinae and Xiphozelinae; the ichneumonids—Acaenitinae, Agriotypinae, Clasinae, Cylloceriinae, Diacritinae, Eucerotinae, Hybrizontinae, Masoninae, Microleptinae, Nesomesochorinae, Oxytorinae, Pedunculinae, Sisyrostolinae and Tatogastrinae. (Phylogeny hand-drawn based upon a consensus from [4,6,7,8,9]).

**Figure 2 biology-10-00050-f002:**
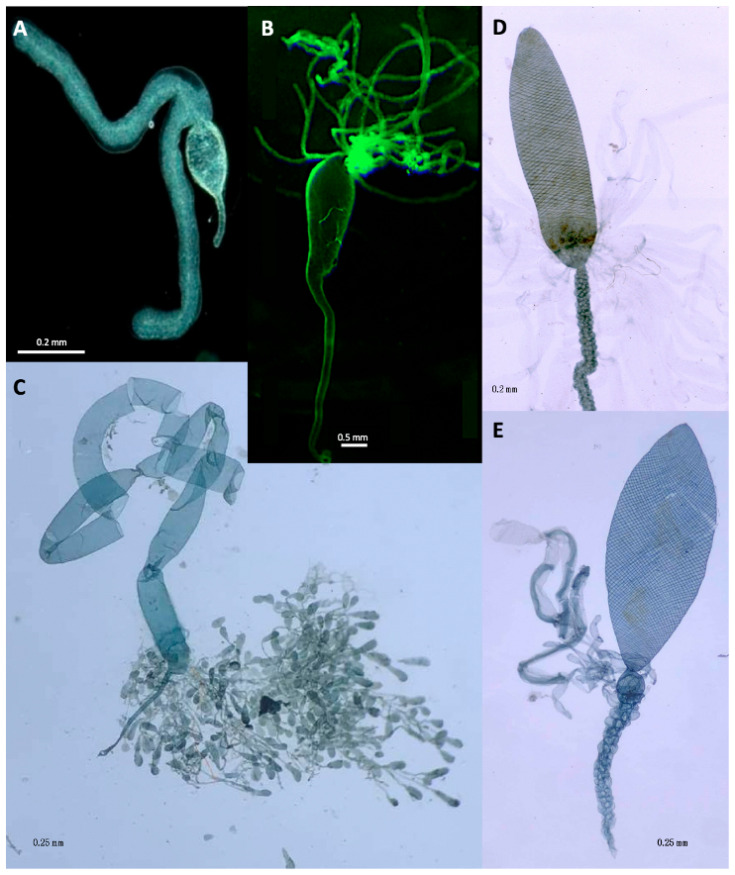
Dissected venom glands and reservoirs or their chitinous intima. (**A**), dissected venom glands and reservoir of *Diadromus collaris* (Ichneumonidae: Ichneumoninae) from [56] (Reproduced under the terms of Creative Commons Attribution Licence CC-BY 4.0 via *Scientific Reports*), scale bar: 0.2 mm; (**B**), dissected venom glands and reservoir of *Pimpla turionellae* from [57] (Reproduced under the terms of Creative Commons Attribution Licence CC-BY 4.0 via *Toxins*), scale bar: 0.5 mm; (**C**–**E**), chlorozol black dyed reservoir, gland and primary duct chitinous intima: (**C**), gen. sp. (Doryctinae), scale bar: 0.25 mm; (**D**), *Bracon* sp. (Braconinae), scale bar: 0.2 mm; (**E**), *Iphiaulax*, unidentified species (Braconinae), scale bar: 0.25 mm.

**Figure 3 biology-10-00050-f003:**
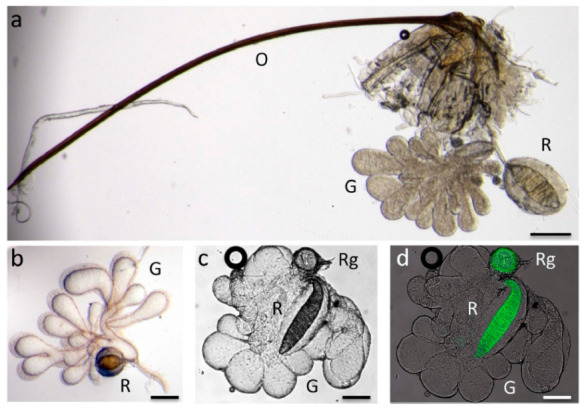
Venom apparatus of two *Psyttalia* species: (**a**) *P. lounsburyi* female venom apparatus composed of a multi-lobed gland (G), a reservoir (R) and a long ovipositor (O); (**b**) Dissected *P. lounsburyi* venom gland showing the thick tissue envelope of the gland and the basal lateral branching of the reservoir; (**c**) *P. concolor* venom apparatus evidencing the small round gland at the base of the apparatus (Rg); (**d**) the same, overlaid with a fluorescence micrograph showing the green auto-fluorescence of the internal spirals of the reservoir and the small round gland. Bars = 100 μm. (Source: from [55] reproduced under the terms of Creative Commons Attribution Licence CC-BY 4.0 via *Scientific Reports*).

**Figure 4 biology-10-00050-f004:**
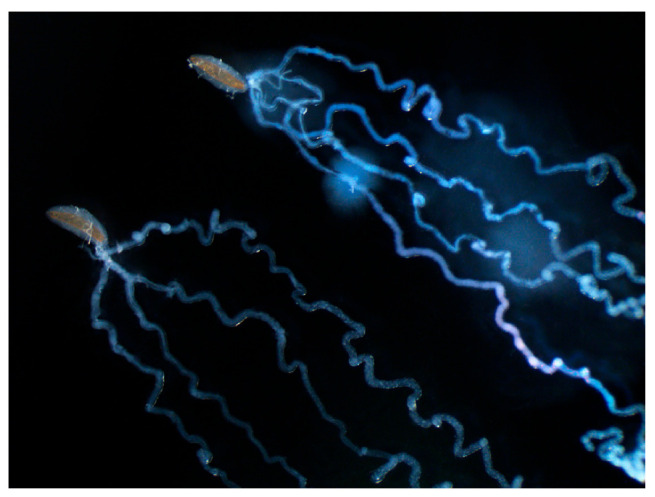
*Diachasmimorpha longicaudata* venom glands with abundant entomopox virus (DlEVP) in their lumens, and escape of virions into surrounding medium from puncture of lower filament in upper right venom apparatus. (Source: courtesy of Kelsey Anne Coffman, Univerity of Georgia, Athens, GA, USA).

**Figure 5 biology-10-00050-f005:**
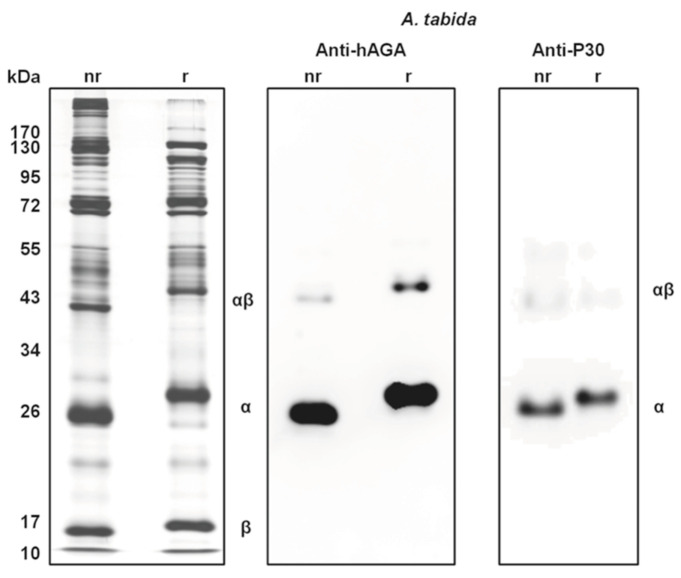
SDS-PAGE separation of *A*. *tabida* venom extracts and asparaginase (AGA) immunostaining. Electrophoretic profiles of venom extracts and immunostaining of AGA under non-reducing conditions (nr) and reducing conditions (r). Gels were silver-stained or analyzed by western blot using the anti-hAGA and anti-P30. The two subunits are indicated *α* and *β*, and the heterodimer by *α**β*. (*A*. *tabida* AGA) antibodies. (Source: from [142] under the terms of Creative Commons Attribution Licence CC-BY 4.0 via *PLoS ONE*).

**Figure 6 biology-10-00050-f006:**
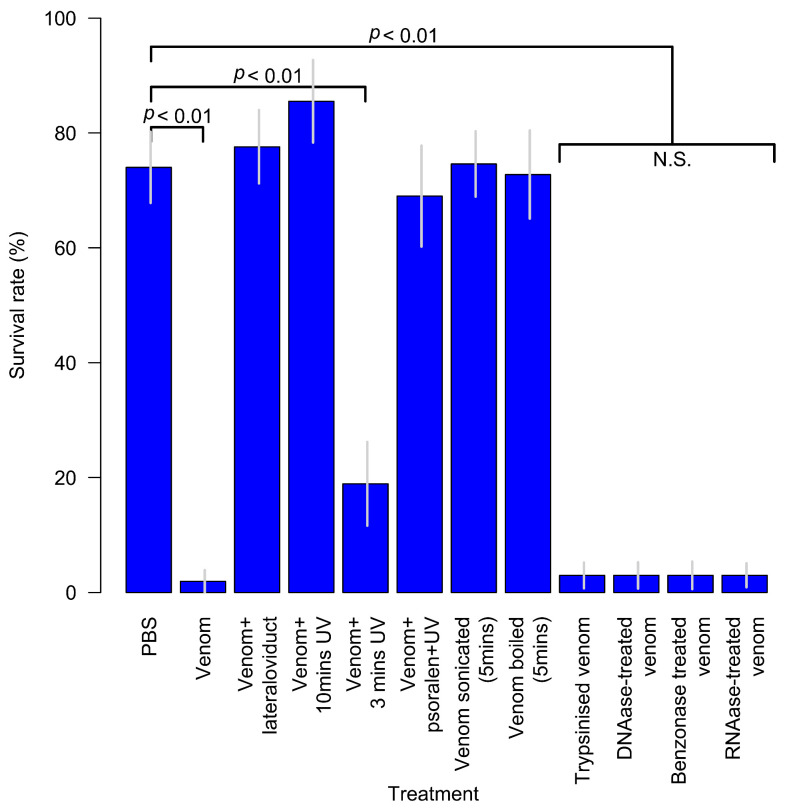
Effects of *Asobara japonica* venom and treated venom on the survivorship of *D. melanogaster* larvae one day after injection. Enzymatic treatments were conducted for 2 h at 25 °C (trypsin) or 37 °C (the others). (Source: redrawn based on [149] under the terms of Creative Commons Attribution Licence CC-BY 4.0 via *PLoS ONE*). N.S.: not significant.

**Figure 7 biology-10-00050-f007:**
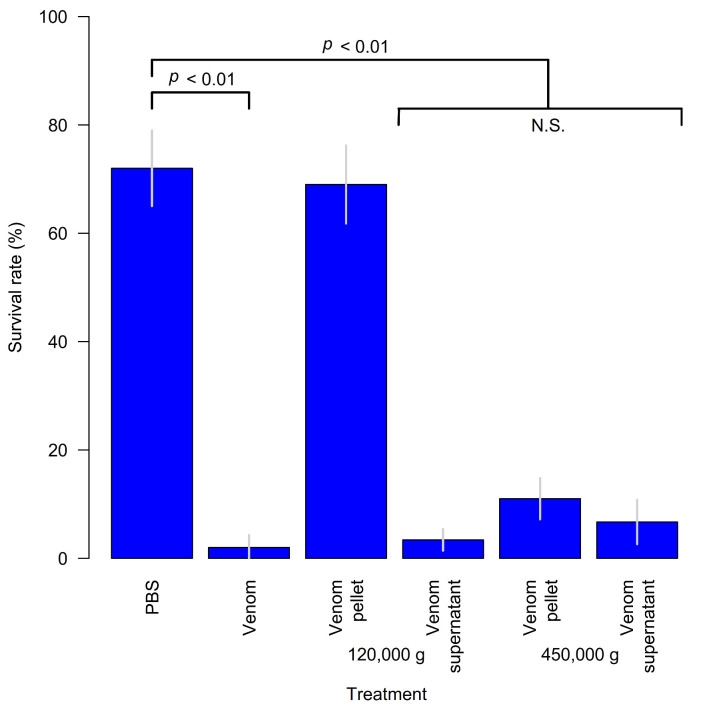
Effects of *Asobara japonica* venom and products of ultracentrifugation on survivorship of *D. melanogaster* larvae one day after injection. (Source: redrawn based on [149] under the terms of Creative Commons Attribution Licence CC-BY 4.0 via *PLoS ONE*). N.S.: not significant.

**Figure 8 biology-10-00050-f008:**
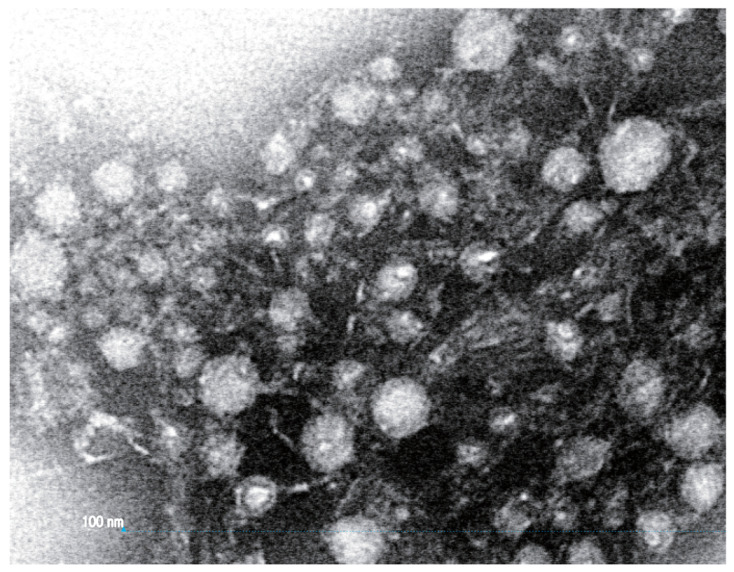
TEM section of pellet obtained by ultracentrifugation of *Asobara japonica* venom at 450,000 g. (Source: from [149] under the terms of Creative Commons Attribution Licence CC-BY 4.0 via *PLoS ONE*).

**Figure 9 biology-10-00050-f009:**
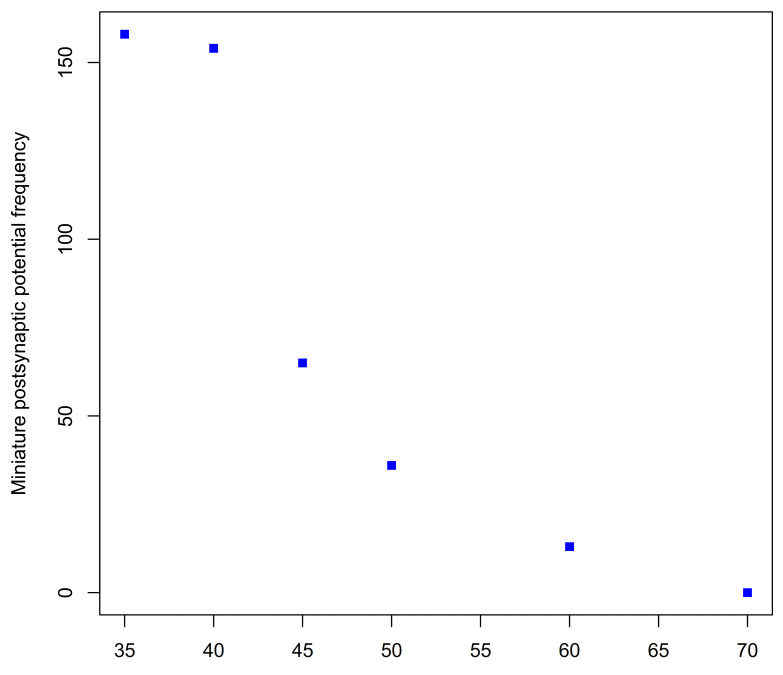
Time course of reduction and loss of miniature parasitism-specific proteins (PSPs) in the silk moth *Philosamia cynthia* following injection of *H. hebetor* venom (Source: data from [184]).

**Figure 10 biology-10-00050-f010:**
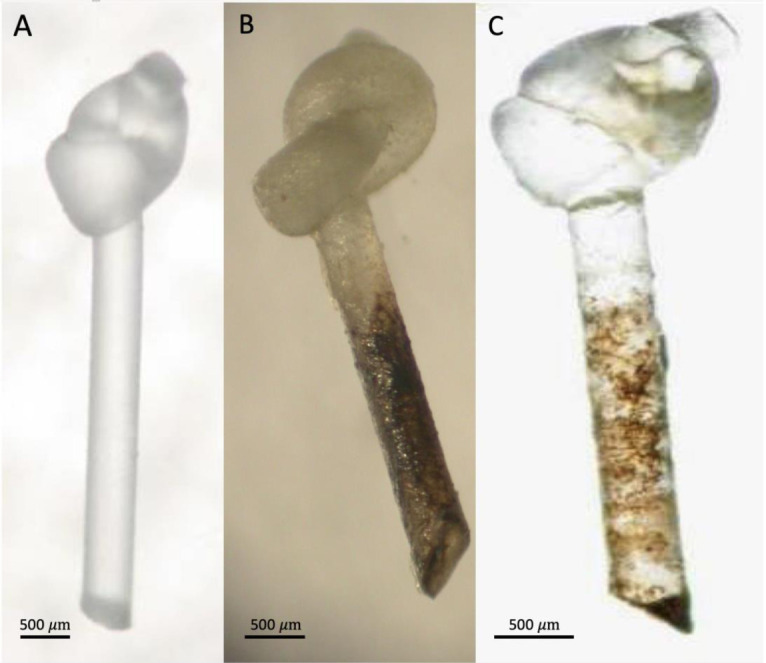
Examples of melanotic encapsulation response to plastic implant by *G. mellonella* larva naturally envenomated by *H. hebetor*. (**A**), initial implant, (**B**), control (untreated) larva 24 h post treatment. (**C**), envenomated larva, 24 hrs post treatment. (Source: reproduced by permission of Ivan Dubovskiy, Novosibirsk State Agrarian University, Russia).

**Figure 11 biology-10-00050-f011:**
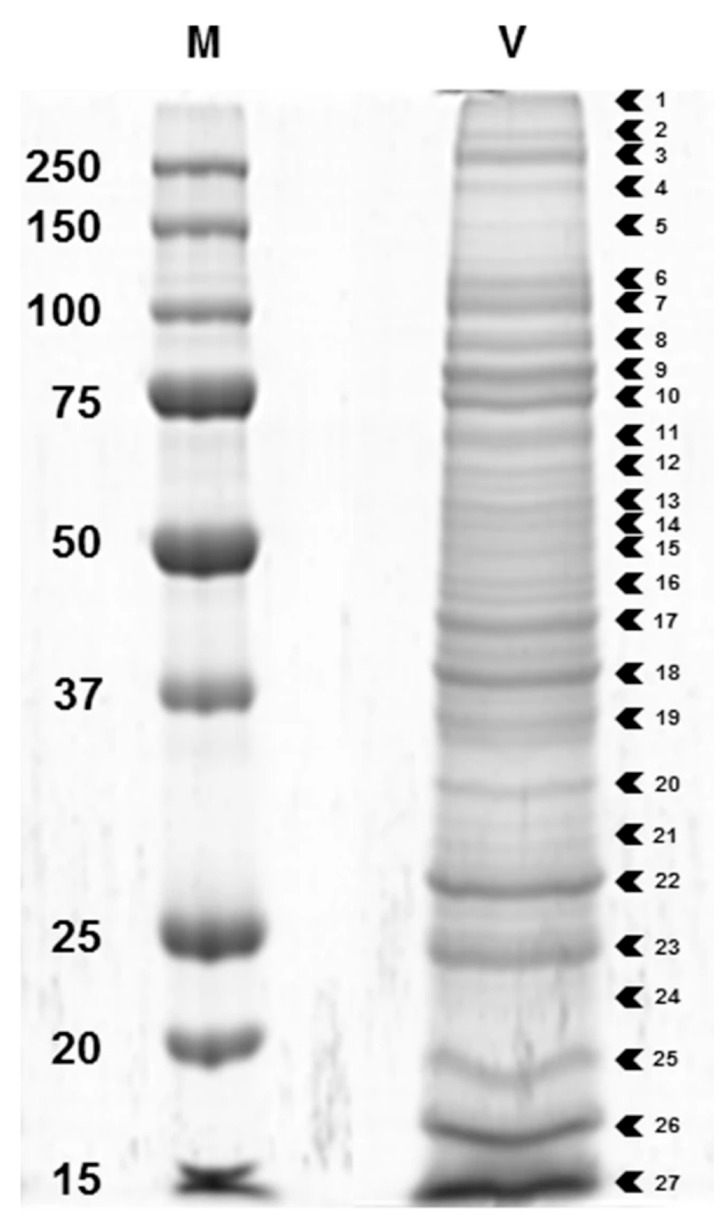
Coomassie Brilliant Blue G250 stained, SDS-PAGE of *H. nigricans* venom extract with known molecular weight protein ladder on left, showing numbered protein bands that were analysed separately. Left hand lane (M) shows protein molecular weight markers, right hand lane shows venom (V) proteins. (Source from [51] under the terms of Creative Commons Attribution Licence CC-BY 4.0 via *BMC Genomics*).

**Figure 12 biology-10-00050-f012:**
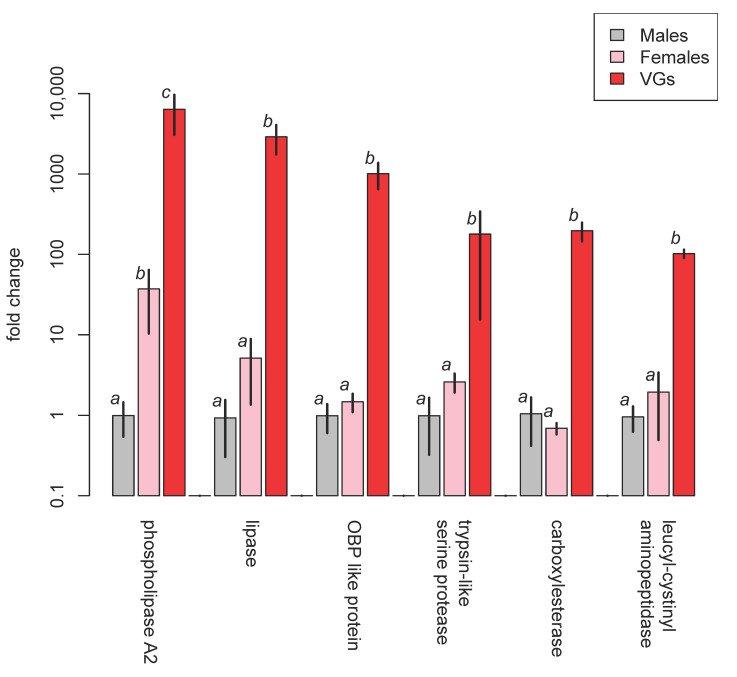
Expression of selected genes in terms of abundance of transcripts measured by qRT-PCR in *H. nigricans* females with their venom glands removed, whole males, and isolated venom glands. Results are presented as mean fold changes on a logarithmic scale, based of three independent biological replicates. Values are standardised with respect to females that had had their venom glands removed and which were assigned a value of 1. Error bars indicate standard error and letters indicate significant difference adjusted for multiple tests using Tukey’s HSD (Source: data from Figure 4 in [51]).

**Figure 13 biology-10-00050-f013:**
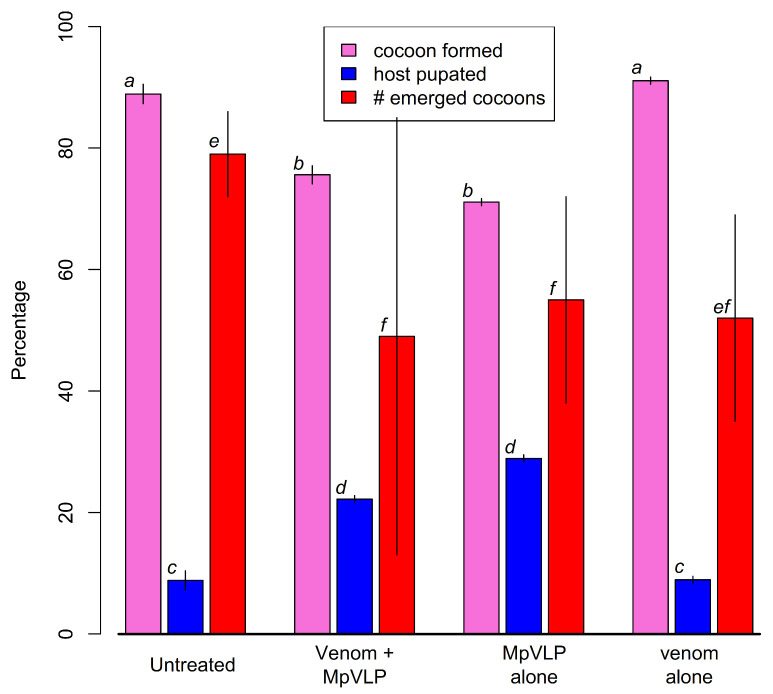
Effects of injection of *M. pulchricornis* products into northern armyworm, *Mythimna separata* (Noctuidae) caterpillars parasitised (on the same day) by *Cotesia kariyai*. Different letters above columns indicate significantly different values at 5% level using Tukey–Kramer test. (Data from Table 2B in [205]).

**Figure 14 biology-10-00050-f014:**
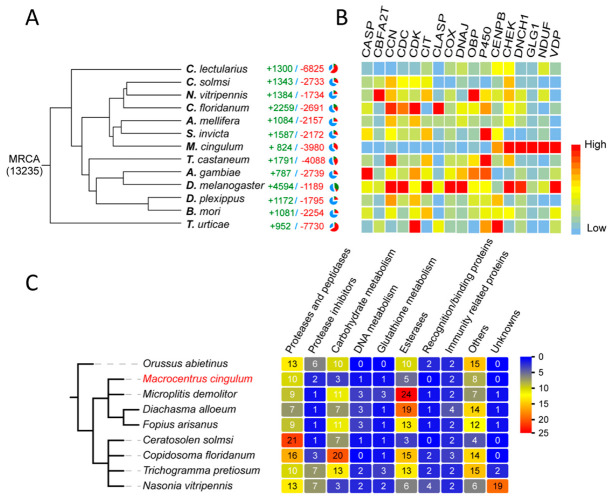
Gene gains and losses of parasitism-associated gene families of *Macrocentrus cingulum* (Braconidae: Macrocentrinae) and a range of other hymenopterans. (**A**) Number of gene families with apparent expansion/contraction among M. cingulum and 12 other species. (**B**) Gene families with significant contraction or expansion. (**C**) Numbers of venom proteins in different parasitic wasps. (Source: from [54] reproduced under the terms of Creative Commons Attribution Licence CC-BY 4.0 via *BMC Genomics*).

**Figure 15 biology-10-00050-f015:**
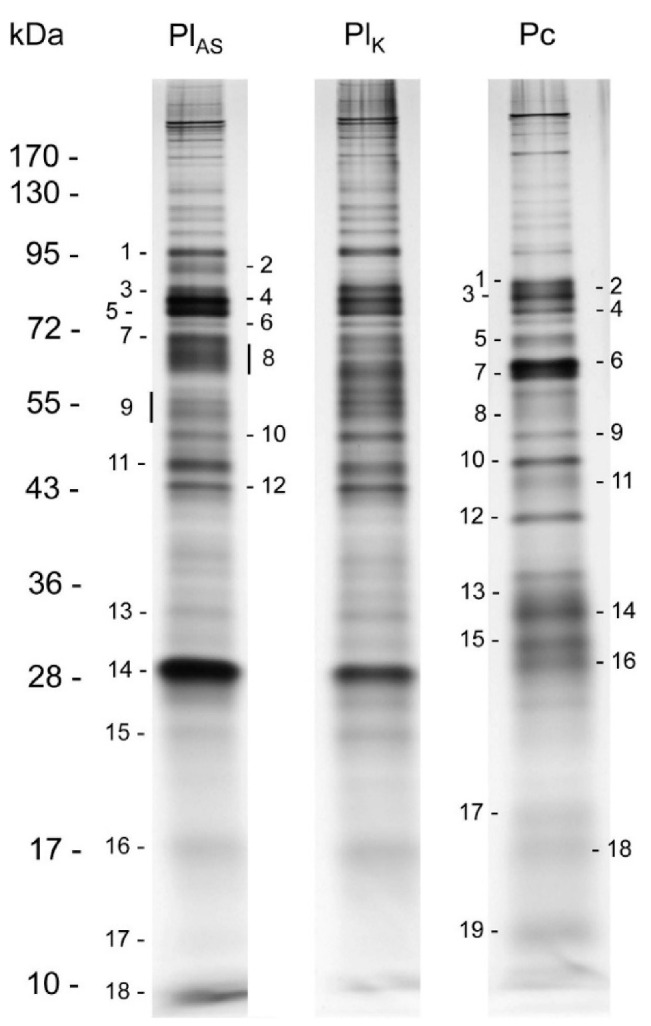
1D SDS-PAGE separation of *P. lounsburyi* (Pl_AS_ and Pl_K_ strains from Kenya and South Africa, respectively) and *P. concolor* venom proteins under reducing conditions and silver staining. Stained protein bands (numbered) excised and submitted for protein identification by LC-MS-MS. (Source: from [55] reproduced under the terms of Creative Commons Attribution Licence CC-BY 4.0 via *Scientific Reports*).

**Figure 16 biology-10-00050-f016:**
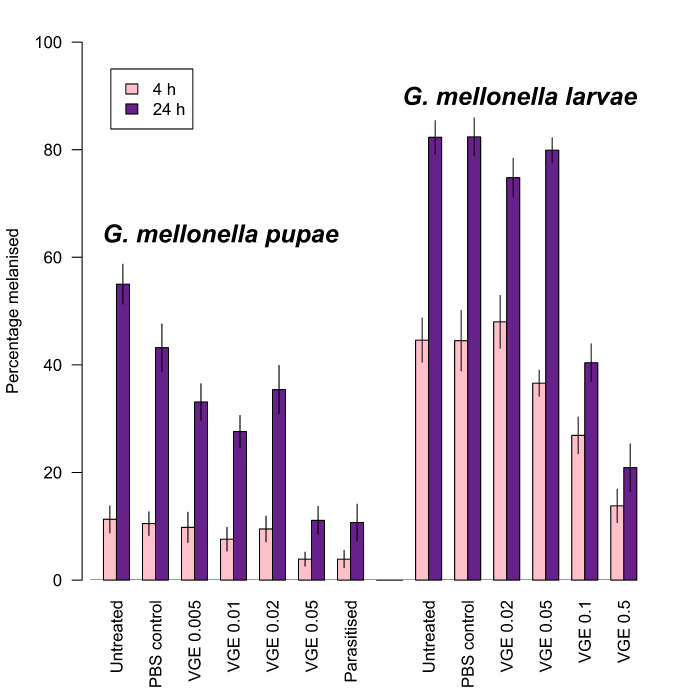
Melanisation of Sephadex DEAE A-25 beads in *Galleria mellonella* pupae and larvae experimentally envenomated and parasitised by *P. turionellae* after 4 and 24 h. (Source: data from Tables 4 and 5 [246]).

**Figure 17 biology-10-00050-f017:**
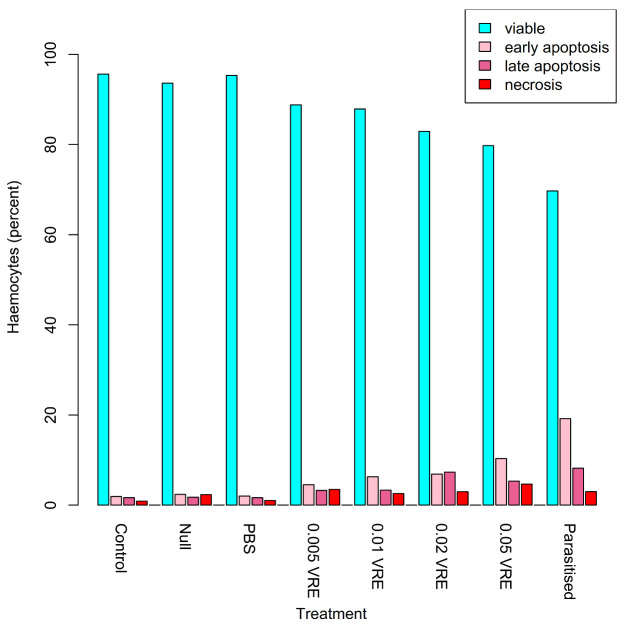
Effect of *P. turionellae* venom and parasitisation on host, *Galleria mellonella* haemocyte health and viability (Source: data from [250]).

**Figure 18 biology-10-00050-f018:**
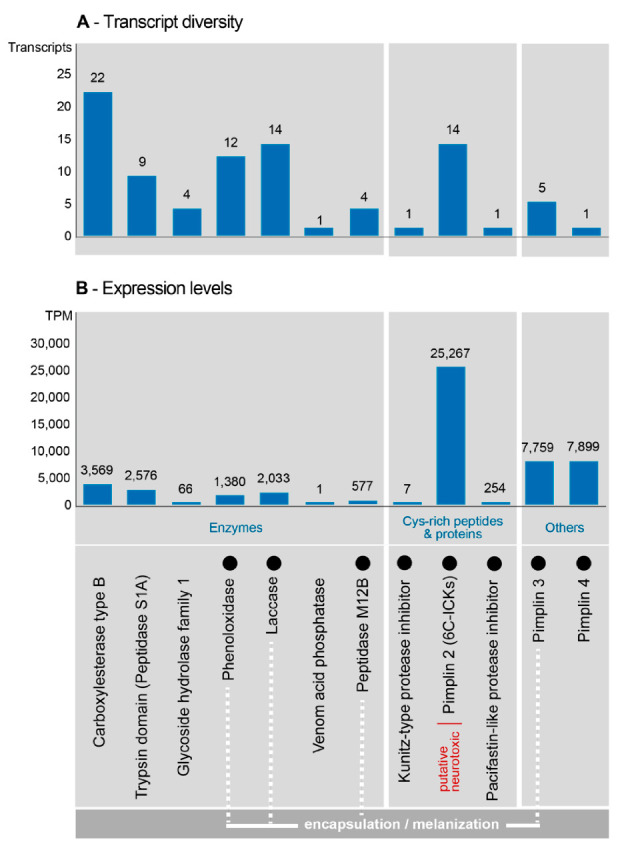
Transcript diversity and expression levels of identified known venom protein families (Source: from [57], reproduced under the terms of Creative Commons Attribution Licence CC-BY 4.0 via *Toxins*).

**Table 1 biology-10-00050-t001:** Summary of transcriptomic and proteomic studies of non-polydnavirus (PDV) ichneumonoid venoms.

Family	Subfamily	Species	Method and Comments	References
Braconidae	Aphidiinae	*Aphidius ervi*	proteomics	[48]
—	Braconinae	*Habrobracon hebetor*	*de novo* sequencing and transcriptome	[49,50]
—	—	*Habrobracon nigricans*	proteo-transcriptomics	[51]
	Euphorinae	*Meteorus puchricornis*	VG transcriptomics and RNA interference	[52]
—	—	*Microctonus hyperodae* & *M. aethioiodes*	transcriptomics and cloning, and pysosequencing of cDNA	[53]
—	Macrocentrinae	*Macrocentrus cingulum*	genome and whole body transcriptome sequencing	[54]
—	Opiinae	*Psyttalia concolor* & *P. lounsburyi*		[55]
Ichneumonidae	Ichneumoninae	*Diadromus collaris*	transcriptomes of VGs and of bodies without VGs	[56]
—	Pimplinae	*Pimpla turionellae*	VG transcriptome	[57]
—	Rhyssinae	*Megarhyssa greenei* & *M. macrurus*	transcriptome of terminal three metasomal segments; venom proteomics using LC-ESI-MS/MS	[58]

**Table 2 biology-10-00050-t002:** Scientific binomens often used in the venomological literature and their corresponding current taxonomically correct versions.

Family	Subfamily	Previous (Incorrect) Combinations	Correct Scientific Name
Braconidae	Alysiinae	*Phaenocarpa persimilis*	*Asobara persimilis*
—	Braconinae	*Bracon brevicornis*	*Habrobracon brevicornis*
—	—	*Bracon hebetor*	*Habrobracon hebetor*
—	—	*Microbracon gelechiae*	*Habrobracon gelechiae*
—	—	*Microbracon hebetor*	*Habrobracon hebetor*
—	—	*Bracon nigricans*	*Habrobracon nigricans*
—	Opiinae	*Biosteres longicaudatus*	*Diachasmimorpha longicaudata*
—	—	*Opius concolor*	*Psyttalia concolor*
Ichneumonidae	Campopleginae	*Nemeritis canescens*	*Venturia canescens*
—	—	*Coccygomimus turionellae*	*Pimpla turionellae*
—	Pimplinae	*Pimpla hypochondriaca*	*Pimpla rufipes*

**Table 3 biology-10-00050-t003:** Putative venom protein components of *Aphidius ervi*. (Source: data from [48]).

Protein	No. of ESTs
Elongation factor 2	6
Endoplasmin	19
γ-glutamyl transpeptidase	539
Leucine rich repeat domain-containing protein	30
Serine protease homologue	97
Serpin	26

**Table 4 biology-10-00050-t004:** LD50s of two *Habrobracon hebetor* venom proteins against various target insects (Source: data from [170]).

Target Insect	Toxin LD_50_ Per Target Weight (μg/g)	
	Brh-I	Brh-V
*Galleria mellonella*	0.0023	0.0001
*Manduca sexta*	0.05	0.04
*Spodoptera exigua*	0.033	0.051
*Heliothis virescens*	0.18	0.26
*Heliothis zea*	0.045	0.085
*Trichoplusia ni*	0.019	0.0038

**Table 5 biology-10-00050-t005:** Putative identifications of venom proteins of *H. nigricans* based of reference to the SwissProt database (https://www.uniprot.org) and InterProScan (https://www.ebi.ac.uk/interpro/) which classifies proteins into functional families. In addition, proteins located in bands 7, 9, 10, 11 showed no similarity to anything on SwissProt and were indicated as protein domain family DUF4803 by InterProScan. Braconidae matches are in bold. (Source: based on Table 2 [51]).

Gel Band	Best Identifications		Venom Proteins Found in other Parasitoid Wasps (Braconids Highlighted) *
	SwissProt	InterProScan	
28	none	odorant-binding protein	Ac, **Ci**, Nv, Lh, Pp
26	phospholipase A2	phospholipase A2	**Hh**, Eo, **Pc**, Tn
27	none	prokaryotic membrane lipoprotein lipid attachment site	**Hh**
11, 12	venom carboxylesterase-6 (*Apis mellifera*)	carboxylesterase, type B	Ac, **Dc**, Hd, Nv
23	venom allergen 5 (*Vespa crabro*)	venom allergen 5-like	**Ci**, Hd, Lh, **Mh**, Nv, Tb
20	chymotrypsin-1 (*Anopheles gambiae*)	serine protease, peptidase, chymotrypsin	**Ae**, **Hh**, **Ci**, **Cr**, Es, Hd, Nv, Pr, Pp, **Tn**
27	none	odorant-binding protein	Ac, **Ci**, Nv, Lh, Pp
6, 7	aminopeptidase M1-A	aminopeptidase N-type	**Cc**, Lb, Pr, **Pl**
13	protein disulfide-isomerase	disulfide isomerase—PDI	**Ae**, **Cc**, De, **Pc**, **Pl**, Pp
13	platelet glycoprotein V	leucine-rich repeat	Ae, **Pc**, **Pl**
16	lipase 3	lipase	**Ci**, Lb, **Ma**, Nv, Ot, Pr, Pp
5	lysosomal alpha-mannosidase	alpha mannosidase	De

* Ac, *Anisopteromalus calandrae*. Ae, *Aphidius ervi*. Cc, *Cotesia chilonis*. Ci, *Chelonus inanitus*. Cr, *Co. rubecula*. Dc, *Dinocampus coccinellae*. De, *Diversinervus elegans*. Eo, *Eupelmus orientalis*. Es, *Euplectrus separatae*. Hd, *Hyposoter didymator*. Hh, *Habrobracon hebetor*. Lb, *Leptopilina boulardi*. Lh, *L. heterotoma*. Ma, *Microctonus aethiopoides*. Mh, *M. hyperodae*. Nv, *Nasonia vitripennis*. Ot, *Ooencyrtus telenomicida*. Pc, *Psyttalia concolor*. Pp, *Pteromalus puparum*. Pl, *Psyttalia lounsburyi*. Pr, *Pimpla rufipes* (= *hypochondriaca*). Tb, *Tetrastichus brontispae*. Tn, *Toxoneuron nigriceps*.

**Table 6 biology-10-00050-t006:** Putative functions of selected cDNA gene clusters from venom gland (VG) filaments of *Meteorus pulchricornis* (Source: based on [52]).

Putative Function	Homology (Top Hits)		Estimated No. in Cluster
	BLAST2GO	BLAST	
cell adhesion/fusion	none	laminin	10
—	CD63 antigen		3
apoptosis induction/cell disruption	proteasome assembly chaperone 2		1
—	none		1
cell motility	none	rabaptin	3
—	cytoplasmic fmr1-interacting protein		1
—	Ras-like GTP-binding protein rho1 isoform x2		1
immune suppression	none	macroglobulin/complement-like protein	25
—	none	serine protease inhibitor	1
—	none	serine protease inhibitor	1
functional venom protein/enzyme	none	haemolysin-like	16
—	none	chitinase	5
—	hyaluronidase	none	4
—	bvpp41b protein	none	1
not determined	none	matrilin	22
—	none	none	21
—	none	methyl-accepting chemotaxis protein	9
—	none	iron transporter	8
—	none	ferrodoxin	7
—	none	none	5

**Table 7 biology-10-00050-t007:** The 14 most abundant RNA transcripts from *Microctonus aethiopoides* VGs (Source: data from Table 5 in [53]).

*M. aethiopoides* Contig Sequence Reads	Putative Function
1651	Lipase based on automated gene annotation
940	Low density lipoprotein receptor involved in transmembrane lipid transport
614	Uncertain: could be an autophosphorylating protein tyrosine kinase
611	Neutral endopeptidase
416	Histidine acid phosphatase based on automated gene annotation
296	Protein involved in cell adhesion and motility
253	Uncertain: some resemblance to snake toxins; allergenic to humans
213	Uncertain: highly glycosilated protein; allergenic to humans
183	Intracellular Ca^2+^-binding protein
155	Fe^+^-binding protein
96	Resembles a gene involved in suppression of apoptosis
94	Lysosomal thiol reductase
93	Lysosome associated protease
91	Chitinase

**Table 8 biology-10-00050-t008:** Putative *Psyttalia* venom proteins with putative functions or present in both species; ranked based on Reads Per Kilobase of transcript, per Million mapped reads (RPKM) values for *P. lounsburyi*, and showing their occurrence in other parasitoid wasps. Braconidae matches are in bold. (Source: data combined from Tables 1 and 2 in [55]).

RPKM (Rank)		Putative Function	Proteins Found in other Parasitoid Wasp Venoms (Braconids Highlighted) *
*P. lounsburyi*	*P. concolor*		
574.51 (9)	3110.46 (1)	DUF4803 domain-containing protein	**Ci**, **Hn**, **Md**, **Mh**
1963.84 (1)	1583.31 (3)	—	
1865.04 (2)	—	Leucine-rich repeat protein	**Ae**, **Hn**, Mg
738.87 (5)	441.51 (12)	DUF4803 domain-containing protein	**Ci**, **Hn**, **Md**, **Mh**
691.22 (6)	947.92 (10)	DUF4803 domain-containing protein	**Ci**, **Hn**, **Md**, **Mh**
627.83 (8)	1178.46 (8)	Neprilysin-like metalloprotease	**Ae**, **Mh**
476.04 (10)	441.52 (12)	DUF4803 domain-containing protein	**Ci**, **Hn**, **Md**, **Mh**
272.37 (11)	1196.52 (7)	GH1 β-glucosidase	**Ae**, **Md**
263.42 (12)	236.32 (17)	Calreticulin	**Hh**, **Mh**
242.5 (13)	1507.73 (4)	—	
186.32 (14)	—	Reprolysin-like metalloprotease	**Ci**, **Md**
143.39 (15)	—		
135.9 (16)	—	Esterase/lipase-like	**Ci**, **Hn**, **Md**
115.21 (17)	1178.46 (8)	Neprilysin-like metalloprotease	**Ae**, **Mh**
112.79 (18)	259.63 (15)	Protein disulfide isomerase	**Ae**, **Cc**, De, **Hn**, Mg, Mm, Pp
85.45 (19)	227.98 (18)	Heat shock protein 70	**Ae**
63.77 (20)	—	Protein disulfide isomerase	**Ae**, **Cc**, De, **Hn**, Mg, Mm, Pp
40.85 (21)	—	Endoplasmin	**Ae**
35.95 (22)	—	DUF4803 domain-containing protein	**Ci**, **Md**, **Mh**
22.74 (23)	199.42 (19)	Protein disulfide isomerase	**Ae**, **Cc**, De, **Hn**, Mg, Mm, Pp
21.08 (24)	—	Puromycin-sensitive aminopeptidase	
17.02 (25)	53.51 (22)	Enolase	
6.13 (26)	—	Arginine kinase-like protein	**Hh**
2.43 (28)	—	Esterase/lipase-like	
2.28 (29)	32.85 (25)	Serpin	**Ae**, **Md**
202 (30)	—	Leucine rich repeat protein	**Ae**, **Hn**, Mg
1.47 (31)	—	Neprilysin-like	**Ae**, **Mh**
1.31 (32)	—	Glycogen phosphorylase	
—	1326.3 (6)	Reprolysin-like metalloprotease	Pr
—	580.77 (11)	Phospholipase A2	Eo, **Hh**, **Hn**, **Md**, Mg, **Tn**
—	1152.41 (9)	DUF4803 domain-containing protein	**Ci**, **Hn**, **Md**, **Mh**
—	360.78 (13)	Annexin	
—	346.13 (14)	Serine carboxypeptidase	**Md**
—	102.94 (20)	Leucine-rich repeat protein	**Ae**, **Hn**
—	60.85 (21)	Protein disulfide isomerase	**Ae**, **Cc**, De, **Hn**, Mg, Mm, Pp
—	39.64 (23)	Leucine-rich repeat protein	**Ae**, **Hn**, Mg
—	36.88 (24)	Ezrin/radixin/moesin family	
—	29.37 (26)	Neprilysin-like metalloprotease	**Ae**, **Mh**
—	12.69 (27)	Aldehyde dehydrogenase	
—	5.03 (28)	Leucine-rich repeat protein	Ae, Hn, Mg
—	4.49 (29)	Leucine-rich repeat protein	Ae, Hn, Mg
—	2.31 (30)	Adenosylhomocysteinase	

* Ac, *Anisopteromalus calandrae*. Ae, *Aphidius ervi*. Ci, *Chelonus inanitus*. Cc, *Cotesia chilonis*. Cr, *Co. rubecula*. De, *Diversinervus elegans*. Eo, *Eupelmus orientalis*. Es, *Euplectrus separatae*. Hd, *Hyposoter didymator*. Hh, *Habrobracon hebetor*. Hn, *Habrobracon nigricans*. Lb, *Leptopilina boulardi*. Lh, *L. heterotoma*. Ma, *Microctonus aethiopoides*. Md, *Microplitis demolitor*. Mh, *M. hyperodae*. Mg, *Megarhyssa greenei*. Mm, *Megarhyssa macrurus*. Nv, *Nasonia vitripennis*. Ot, *Ooencyrtus telenomicida*. Pr, *Pimpla rufipes* (= *hypochondriaca*). Pp, *Pteromalus puparum*. Tb, *Tetrastichus brontispae*. Tn, *Toxoneuron nigriceps*.

**Table 9 biology-10-00050-t009:** Summary of major protein components identified in the venom of *Pimpla rufipes*.

Class	Protein Name	Putative Function	Comments	References
Neurotoxins	Pimplin	major paralytic factor in venom	no similarity to other known proteins	[227]
—	Cys-rich venom protein 3	possible minor neurotoxin	atracotoxin-like	[232]
—	Cys-rich venom protein 5	possible minor neurotoxin	conotoxin-like	[232]
Protease inhibitors	Cys-rich venom protein 1	—	protease inhibitor	[232]
—	Cys-rich venom protein 2	—	Kunitz type protease inhibitor	[232]
—	Cys-rich venom protein 4	—	pacifastin; protease inhibitor	[232]
—	Cys-rich venom protein 6	—	protease inhibitor	[232]
Other enzymes	Acid phosphatase	—	possibly related to purine release	[233]
—	Laccase, lac1, Phlac	oxidation	—	[234]
—	tre1	—	similar to trehalase	[234]
—	Phenoloxidase I	putative haemocyte disruption	from VG cDNA library	[230]
—	Phenoloxidase II	putative haemocyte disruption	from VG cDNA library	[230]
—	Phenoloxidase III	putative haemocyte disruption	from VG cDNA library	[230]
—	Metalloprotease	—	similar to snake venom reprolysin-type metalloproteases	[235]
—	Serine protease	—	—	[236]
Haemocyte anti aggregation proteins	VPr1	probably reduces host encapsulation ability	haemocyte inactivation	[224,225,226,234]
—	VPr3	hydrolysis	Antihaemocyte aggregation	[224,237]
others				[228]

**Table 10 biology-10-00050-t010:** Named venom proteins identified in the venom of *P. turionellae* and attributes. (Source adapted from [57] under the terms of Creative Commons Attribution Licence CC-BY 4.0 via *MDPI Toxins*).

Protein	Transcripts Per Million	Amino Acid Length	Scaffold	Putative Function/Comments
pimplin2	25,267	64–115	X-CX_7_ -[C-X_6_ -C-X_5–8_-CC-X_2–4_ -C-X_6–9_]-X	possibly similar neurotoxic asilid1 sequences from robber flies though the Cys-scaffold is different to a typical ICK one
pimplin3	7759	167–315	potential P and C scaffold	same family as venom protein1, Vpr1 from *P. rufipes*
pimplin4	7899	70–78	no cysteine scaffold, 3 P residues	same family as small venom protein2, svp2 from *P. rufipes*

**Table 11 biology-10-00050-t011:** Selected non-PDV carrying ichneumonoid wasp species not yet investigated but which ought to be amenable to culture and laboratory investigation.

Family	Subfamily	Species	Host(s)	Notes	Reference
Braconidae	Agathidinae	*Alabagrus stigma*	*Diatraea saccharalis*	was cultured at Texas A&M in the 1980s by James Smith’s group	Nothing appears to have been published
—	Alysiinae	*Aphaereta pallipes*	onion maggot, *Delia antiqua*	—	[258]
—	Doryctinae	*Allorhogas pyralophagus*	*Eoreuma loftini*	gregarious idiobiont ectoparasitoid	[259,260]
—	—	*Heterospilus prosopidis*	various bruchid beetle larvae feeding within various beans	easily cultured, cyclostome idiobiont ectoparasitoid	
—	—	*Spathius exarator*	common furniture beetle, *Anobium punctatum*	idiobiont ectoparasitoid	[261]
—	Euphorinae	*Dinocamptus coccinellae*	many species of aphidophagous ladybird beetles (Coccinellidae)	cosmopolitan endoparasitoid	[262]
—	Orgilinae	*Orgilus lepidus*	potato tuber moth. *Phthorimaea operculella*	koinobiont larval endoparasitoid; sometimes sold commercially	[263,264]
Ichneumonidae	Anomaloninae	*Heteropelma scaposum*	*Helicoverpa* spp. (Lepidoptera: Noctuidae)	solitary koinobiont endoparasitoid	[265]
—	Cremastinae	*Pristomerus vulnerator*	codling moth, *Cydia pomonella* (Lepidoptera: Tortricidae)	koinobiont larval endoparasitoid	
—	Cryptinae	*Diapetimorpha introita*	*Spodoptera frugiperda* (Lepidoptera: Noctuidae)	pupal ectoparasitoid; can be reared on artificiual diet	[266,267]
—	—	*Agrothereutes lanceolatus*	various tortricids and pyralids, including *Chilo suppressalis*, *Glyphodes pyloalis* and *Homona magnanima*	solitary idiobiont ectoparasitoid of prepupal (and early pupal) host	[268]
—	—	*Hemiteles graculus*	alfalfa weevil, *Hypera positica*	solitary idiobiont ectoparasitoid of prepupal and pupal host	[269]
—	—	*Mastrus ridens*	codling moth, *Cydia pomonella*	gregarious idiobiont ectoparasitoid of cocooned mature larval host	[270]
—	—	*Mallochia pyralidis*	*Eoreuma loftini* (Pyralidae)	idiobiont ectoparasitoid	[271]
—	Pimplinae	*Itoplectis naranyae*	include *Galleria mellonella, Chilo suppressalis*	pupal endoparasitoid	
—	—	*Exeristes roborator*	include *Galleria mellonella*, *Pectinophora gossypiella*, *Phthorimaea operculella*, *Larinus sturnus*	highly polyphagous endoparasitoid	[272]
—	Tryphoninae	*Netelia producta*	*Helicoverpa* spp. (Lepidoptera: Noctuidae)	koinobiont larval ectoparasitoid	[265]

## Data Availability

Data sharing not applicable.

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
