# Peer review of "Review of Venoms of Non-Polydnavirus Carrying Ichneumonoid Wasps"

_biology, 2021, doi:10.3390/biology10010050_

Round 1
Reviewer 1 Report
No further comments.
Author Response
Nothing to respond
Reviewer 2 Report
The authors have done a good job of revising this manuscript and I suggest it is now ready for publication.
Author Response
Nothing to respond
Reviewer 3 Report
Title: Review of venoms of non-polydnavirus carrying ichneumonoid wasps
Although the authors have made corrections and improved for the typing errors, and I understand that they have done an important work on the literature, I still find this review too long with a lack of synthesis.
Minor comments
Line 37-38 even in non PDV wasps the ovarian fluid can have some important effect (i.e. A. japonica).
line 169 remove "of peptides" (here, it is mainly for proteins)
line 188-189 it seems to me that the word "reservoir" is missing before thick musculature
Fig 2 line 222 where is panel F ?
line 228-230 please provide a ref for this work
line 263 to 266 sentences are redondant
some minor typos to correct
Author Response
Reply to Biology (3)
Line 37-38 even in non PDV wasps the ovarian fluid can have some important effect (i.e. A. japonica). < We have added mention of this
line 169 remove "of peptides" (here, it is mainly for proteins) < We have changed this
line 188-189 it seems to me that the word "reservoir" is missing before thick musculature. < We have amended this
Fig 2 line 222 where is panel F ? < We have corrected the legend lettering
line 228-230 please provide a ref for this work < The works ARE referenced but the font was small for [70]; we have corrected the font
line 263 to 266 sentences are redondant <- We have deleted one redundant sentence and modified adjacent wording appropriately
___________________________________________________________________
- With regard to referee 3's criticism of the length of the review, we disagree and note that: referee 1 saw this more positively stating "... a remarkably comprehensive, well-illustrated review of venoms of non-polydnavirus carrying ichneumonoid wasps"
- referee 2 wrote "This manuscript is an in-depth review of the venom systems of those ichneumonoidean wasps that do not carry polyDNAviruses. The authors have assembled a very large amount of data and applied critical analysis to it which is for the most part high quality."
With regard to referee 3's criticism of lack of synthesis, we believe we have done all that is reasonably possible given the relative paucity of taxa studied and methods employed, the marked interspecific differences in mode of action and, apparently, in biochemical mechanisms involved. Much of the purpose of a review is to draw attention to areas warranting further study (which we do), and not just synthesis which is only possible with far better and more evenly studied systems.
In summary we think our MS is a useful, in depth, review that points out what synthesis is reasonable at present and indicates areas were additional methods and taxa would be beneficial.
___________________________________________________________________
line 957 we have indented the paragraph first line

This manuscript is a resubmission of an earlier submission. The following is a list of the peer review reports and author responses from that submission.
Round 1
Reviewer 1 Report
Quicke and Butcher present a remarkably comprehensive, well-illustrated review of venoms of non-polydnavirus carrying ichneumonoid wasps.
The manuscript is very detailed, covering the difficulties of maintaining and continuing wasp populations to study, the various ways wasps are classified depending on the type of scientist studying them, the temporary vs. permanent types of neurotoxins used, the strategies of immobilizing prey and leaving their eggs/larvae within the host, etc.
The work then goes on for a couple of dozen pages describing the genomics and proteomics of these various wasp families, with a focus on venom constituents and venom gland/injection mechanisms. The work contains occasional taxonomy as well to compare and contrast between the wasps.
The conclusions drawn include that very little has been done so far with respect to investigation of the venoms of these diverse species, and the effects of the venom should be a major focus.
This manuscript is a hard read, longer than many chapters I have reviewed. It has twelve pages of references alone. Nevertheless, it serves its purpose of documenting these wasps in a comprehensive manner.
The paper suffers from numerous spelling and syntax errors – the authors should make an effort to attend to this significant weakness.
Beyond this, I have no important comments.
Author Response
Dear Reviewer 1,
Quicke and Butcher present a remarkably comprehensive, well-illustrated review of venoms of non-polydnavirus carrying ichneumonoid wasps.
The manuscript is very detailed, covering the difficulties of maintaining and continuing wasp populations to study, the various ways wasps are classified depending on the type of scientist studying them, the temporary vs. permanent types of neurotoxins used, the strategies of immobilizing prey and leaving their eggs/larvae within the host, etc.
The work then goes on for a couple of dozen pages describing the genomics and proteomics of these various wasp families, with a focus on venom constituents and venom gland/injection mechanisms. The work contains occasional taxonomy as well to compare and contrast between the wasps.
The conclusions drawn include that very little has been done so far with respect to investigation of the venoms of these diverse species, and the effects of the venom should be a major focus.
This manuscript is a hard read, longer than many chapters I have reviewed. It has twelve pages of references alone. Nevertheless, it serves its purpose of documenting these wasps in a comprehensive manner.
The paper suffers from numerous spelling and syntax errors – the authors should make an effort to attend to this significant weakness.
DONE, we have thoroughly checked the spelling ourselves as well as correcting typos pointed out by the other two reviewers.
Best regards,
Donald and Buntika
Reviewer 2 Report
Review of biology-99628-peer-review-v1
This manuscript is an in-depth review of the venom systems of those ichneumonoidean wasps that do not carry polyDNAviruses. The authors have assembled a very large amount of data and applied critical analysis to it which is for the most part high quality. It is well-presented and timely given I am not aware of any very similar recent reviews and the rapidly changing nature of the field due to incorporation of new technologies, and I believe it will represent a valuable resource for researchers with interest in the field of ichneumonoidean venoms. I suggest that it should be published after minor revisions to fix the many minor issues identified below.
Specific comments
- Lines 65–68: I think 'easy' should be 'ease' throughout this sentence
- Figure 1: Seems like the clade of Braconidae + Trachypetidae is labelled Braconidae, could this be repositioned? Also I found the labelling of the first four columns with 'o /' confusing
- Lines 88,119,120,123,128,269,345,361,365,862–868, possibly elsewhere: Italicise binomial names
- Line 92: subfamilies missing a
- Line 104: since referring to other studies, consider citing some here
- Line 106–107: not sure I understand how proteotranscriptomics (as wonderful as it is) negates functional testing of these venoms/toxins (or what exactly was meant here)
- Lines 207-209: having attempted venom gland dissections of similarly sized wasps, I would say it is extremely difficult to separate the different parts of the venom apparatus of a 2 mm wasp in a way that retains their contents without spilling/mixing to collect material for proteomics
- Line 306: tt should be it
- Line 355: not sure what is meant by 'positive dose dependent decrease', perhaps reword?
- Line 356: perhaps 'unknown and could' is better
- Line 396: beneficial
- Line 397: targeted
- Line 473: should this say glutamate-gated ion channels?
- Line 652: not sure how impurities could explain this mass discrepancy
- Line 667: isolates should be isolated
- Figure 10: the white balance between panels B and C is very different, making it hard to compare melanisation. Are these the originals or have they been differentially processed?
- Line 704: indicated
- Table 5: braconids don't seem to be highlighted/bolded as claimed? Just gel band 28
- Line 757: pupaption should be pupation
- Line 779: assembelled should be assembled
- Line 836: mutualistic
- Line 842: ofther should be other
- Line 843: separation
- Table 10: as far as I can tell the entry in the first row is a bit misleading (the paper it cites is a bit misleading). The sequence scaffold shown in the table has an additional Cys after the Cys doublet and is missing one after as compared to a typical ICK. Asilidin1 does not have a solved tertiary structure, though it is likely to be an ICK. The similarity between the wasp and robber fly peptides is based on a big clustering algorithm that does not produce any confidence values and as far as I can tell does not reach the level of BLAST similarity. I would reword the entry after looking again at the primary data
- Line 1000: Reference 245 used only SDS-PAGE to compare these peptides to standards and so the 'identification' of mellitin and apamin in Pimpla venom is very unlikely to be correct, I suggest removing this. As far as I am aware, there are homologues to aculeatoxins such as mellitin in venoms of non-aculeate apocritans but the apamin family is much more phylogenetically restricted
- Line 1114 and possibly elsewhere: ensure consistent abbreviation of journal names
Author Response
Dear Reviewer 2
Thank you very much for your comments. We have done all the corrections as you suggested.
Specific comments
- Lines 65–68: I think 'easy' should be 'ease' throughout this sentence DONE, this came from some automated spell-checker. Sorry!
- Figure 1: Seems like the clade of Braconidae + Trachypetidae is labelled Braconidae, could this be repositioned? Also I found the labelling of the first four columns with 'o /' confusing. DONE, we have adjusted the family labels and changed the o / symbols to coloured squares
- Lines 88,119,120,123,128,269,345,361,365,862–868, possibly elsewhere: Italicise binomial names DONE
- Line 92: subfamilies missing a DONE
- Line 104: since referring to other studies, consider citing some here We do not fully understand what types of study the referee would want cited here, we have not cited any since basically that would be almost all the remaining references in the MS.
- Line 106–107: not sure I understand how proteotranscriptomics (as wonderful as it is) negates functional testing of these venoms/toxins (or what exactly was meant here) We think the sentence is clear – it is referring to the lack of immunological investigations of venom component affinities and not to do with functional testing
- Lines 207-209: having attempted venom gland dissections of similarly sized wasps, I would say it is extremely difficult to separate the different parts of the venom apparatus of a 2 mm wasp in a way that retains their contents without spilling/mixing to collect material for proteomics
- Line 306: tt should be it DONE
- Line 355: not sure what is meant by 'positive dose dependent decrease', perhaps reword? We have deleted 'positive' though think the orifinal was less ambiguous
- Line 356: perhaps 'unknown and could' is better DONE
- Line 396: beneficial DONE
- Line 397: targeted DONE
- Line 473: should this say glutamate-gated ion channels? No, the normal terminology is 'X is a glutamate agonist'; we could write 'X is a potent agonist of glutamate at glutamate-gated ion channels' but that seems excessive.
- Line 652: not sure how impurities could explain this mass discrepancy This is a suggestion in the cited paper, Spanjer et al., who were familiar with the methodologies used at the time
- Line 667: isolates should be isolated DONE
- Figure 10: the white balance between panels B and C is very different, making it hard to compare melanisation. Are these the originals or have they been differentially processed? They were originals from the author. We have replaced the figure with exposure of B adjusted to match that of C. The difference in melanisation is clear. The figure is purely for illustrative purposes as the cited publication used more sophisticated comparative image analysis.
- Line 704: indicated DONE
- Table 5: braconids don't seem to be highlighted/bolded as claimed? Just gel band 28 DONE
- Line 757: pupaption should be pupation. DONE
- Line 779: assembelled should be assembled DONE
- Line 836: mutualistic DONE
- Line 842: ofther should be other DONE
- Line 843: separation DONE
- Table 10: as far as I can tell the entry in the first row is a bit misleading (the paper it cites is a bit misleading). The sequence scaffold shown in the table has an additional Cys after the Cys doublet and is missing one after as compared to a typical ICK. Asilidin1 does not have a solved tertiary structure, though it is likely to be an ICK. The similarity between the wasp and robber fly peptides is based on a big clustering algorithm that does not produce any confidence values and as far as I can tell does not reach the level of BLAST similarity. I would reword the entry after looking again at the primary data We have added a caveat in Table 10
- Line 1000: Reference 245 used only SDS-PAGE to compare these peptides to standards and so the 'identification' of mellitin and apamin in Pimpla venom is very unlikely to be correct, I suggest removing this. As far as I am aware, there are homologues to aculeatoxins such as mellitin in venoms of non-aculeate apocritans but the apamin family is much more phylogenetically restricted. We have modified the text to reflect some uncertainty. However, reference 245 ALSO USED reverse phase HPLC and both apamin and mellitin standards puchased from Sigma.Thus we think that, although based only on co-elution, there is more than just SDS-PAGE as evidence and so have kept this in the review with added caveats.
- Line 1114 and possibly elsewhere: ensure consistent abbreviation of journal names DONE
Best regards,
Donald and Buntika
Reviewer 3 Report
Reviews on the manuscript untitled "Review of venoms of non-polydnavirus carrying ichneumonoid wasps"
This manuscript used mostly data from different sources to present an overview of the recent advances on the composition of venom from parasitoid wasps without polydnaviruses. While some of the species they talked about are clearly without PDVs, for some other this is not yet clearly established or if it is, it is not indicated clearly with referenced works. Overall I found the manuscript is not very good, not well constructed, and contains errors (both in the text and the reported data).Some parts of the text are too long and not focused on the venom of non-PDV venoms from ichneumonoids. The authors have not synthesized the data and only made a catalog without any critical view and sometime extrapolate from the results or are not reporting correctly the data.
I also suggest to the author to read carefully their manuscript to remove the many typing and text errors before submitting it.
The first author has no email address?
More specifically
In the summary they wrote 40,000 species (L15) when in the abstract (L30) and the introduction (L52) it is 100,000 species.
L16 the word venom is certainly missing from the sentence
L21 peptides and not peptines
Polydnaviruses are not true virus, using the word "infect" is not really true, they enter host cells and the wasp DNA they carry are translated into proteins that have profound effects on the cell physiology; PDVs are not replicated in the host, this is not clearly explain and all along the text the authors keep the ambiguity.
All along the text many of the latin name of the species are not in italic .
L99 I suppose that "virus" is certainly missing from the sentence (non-PDVs virus ?)
L105 I don't understand what the author mean
I don't understand the chapter 1.2 particularly since venom from Megarhyssa species have been studied (Pook, V. (2016)) and this study is even listed in table 1
Line 153 - 154 is unclear
L164 mass spectrometry
in table 1 spelling of Meteorus pulchricornis and Psyttalia lounsburyi are wrong; no details are given for psytallia species
L171- I am not sure that this paragraph and table 2 are necessary. It is condescendant toward physiologists to think they are not aware that taxonomists changed the name of the species and then are not up to date. Many websites in different museum or such as the catalogue of life keep tract of these changes that happen for some of them (in table 2) more than 40 years ago.
Line 180-
I think that in this paragraph there is some mistake between the venom apparatus (VG + reservoir) and the venom gland. Both are not interchangeable, and some of the description correspond to the reservoir and not to the gland. This need to be clarified. Moreover I am not convinced that this paragraph is necessary.
Line 199 Feature not Fearure
Line 205-209 Do the authors have any references to support this ?
Line 218 and after. While the figure number and the ref are smaller ?
Line 222, what means macerated ?
Line 237, please explain more clearly what is class 3, not all readers are familiar with the epidermal gland classification.
Line 239-242, This should be address to the authors of the publication, not necessary
paragraph 2.1 The L244 to 249 are not informative and are misleading compare to the title.
While the DIEPV are produced in the venom gland D. Longicaudata, they are not necessary for the parasitism. Several recent publications showed also that other type of symbiotic virus are present in different strains. (Simmonds, T., Carrillo, D., Burke, G. (2016). Characterization of a venom gland-associated rhabdovirus in the parasitoid wasp Diachasmimorpha longicaudata Journal of Insect Physiology 91(), 48-55. https://dx.doi.org/10.1016/j.jinsphys.2016.06.009)
To my knowledge the only species with a clear description of VLPs produced in venom gland is M. pulchricornis. The presence of particles or vesicles may be looking like virus in venom from some species has been reported but not confirmed by other mean than microscopy.
281-283 necessary since VLPs are produced in the ovary in this species?
334-341 I don't understand the rationale of this paragraph
L344 what's mean changes in host biochemical composition ?
L346 the physical appearance of the hemolymph ?
L362-364 this may help the teratocytes growth or to decrease the host defense thus indirectly benefits to the wasp larva
L367 Proteins up-regulated and down-regulated ?
L374, castration is mostly viewed as a competition for ressources between the parasitoid larva and the host reproduction.
L376-380 the publications are mainly demonstrating that GGT is the enzyme responsible for castration.
Part 3.3 there is much more information on venom effects on the host immune system than the few reported here
Part 4; It seems that from here we start a new review on venom?
This is too long before starting the true review on the subject
The text is very long and reports mainly what is in the publication. A synthesis effort should be made, the reader is already lost by the previous parts and don't really understand what you want to talk about. I suggest this review to be shorten since a large part is not directly related to the subject.
Please when you modify a figure from a publication even under CC-BY licence you should indicate it.
Overall although the authors have made some bibliographic efforts a large part of the manuscript is not clear and not focused on the subject.
Author Response
Dear Reviewer 3,
Thank you very much for your comments.
More specifically
In the summary they wrote 40,000 species (L15) when in the abstract (L30) and the introduction (L52) it is 100,000 species. This is correct. 40,000 described species, estimates of total number are far greater
L16 the word venom is certainly missing from the sentence DONE
L21 peptides and not peptines DONE
Polydnaviruses are not true virus, using the word "infect" is not really true, they enter host cells and the wasp DNA they carry are translated into proteins that have profound effects on the cell physiology; PDVs are not replicated in the host, this is not clearly explain and all along the text the authors keep the ambiguity.
All along the text many of the latin name of the species are not in italic . DONE
L99 I suppose that "virus" is certainly missing from the sentence (non-PDVs virus ?) DONE
L105 I don't understand what the author mean
I don't understand the chapter 1.2 particularly since venom from Megarhyssa species have been studied (Pook, V. (2016)) and this study is even listed in table 1
Line 153 - 154 is unclear
L164 mass spectrometry DONE
in table 1 spelling of Meteorus pulchricornis DONE and Psyttalia lounsburyi DONE are wrong; no details are given for psytallia species
L171- I am not sure that this paragraph and table 2 are necessary. It is condescendant toward physiologists to think they are not aware that taxonomists changed the name of the species and then are not up to date. Many websites in different museum or such as the catalogue of life keep tract of these changes that happen for some of them (in table 2) more than 40 years ago. We do not agree about the need for this because this type of review is not meant for experts in the field but for others who may be interested in the topic and not be aware of such name changes. The table is aimed a facilitating their obtaining further information, because for example, if they search only for a name used in physiological literature they may easily miss recent papers on ecology or behaviour or relationships that may be relevant to them. We certainly do not mean it to be condescendant.
Line 180-
I think that in this paragraph there is some mistake between the venom apparatus (VG + reservoir) and the venom gland. Both are not interchangeable, and some of the description correspond to the reservoir and not to the gland. This need to be clarified. Moreover I am not convinced that this paragraph is necessary.
Line 199 Feature not Fearure DONE
Line 205-209 Do the authors have any references to support this ?
Line 218 and after. While the figure number and the ref are smaller ? DONE
Line 222, what means macerated ? This is a standard entomological and microscopical term denoting that soft tissues have been removed by treatment withcaustic agent. We have changed the text to be more specific.
Line 237, please explain more clearly what is class 3, not all readers are familiar with the epidermal gland classification. DONE, we have added a definition as requested.
Line 239-242, This should be address to the authors of the publication, not necessary. We disagree that this should not be mentioned in this review but have modified the wording. There is abundant (hundreds of papers) that describe and illustrate Dufour's glands in Hymenoptera and one of us (DLJQ) has extensive experience in dissecting the female reproductive glands of parasitoid wasps including members of this genus. We have added references to some.
paragraph 2.1 The L244 to 249 are not informative and are misleading compare to the title. We do not understand why this is misleading as it clearly distingushes polydnaviruses from viruses and virus-like particles found in or produced by the venom glands. it seems to us to act as a perfectly reasonable introductory paragraph to this section.
While the DIEPV are produced in the venom gland D. Longicaudata, they are not necessary for the parasitism. Several recent publications showed also that other type of symbiotic virus are present in different strains. (Simmonds, T., Carrillo, D., Burke, G. (2016). Characterization of a venom gland-associated rhabdovirus in the parasitoid wasp Diachasmimorpha longicaudata Journal of Insect Physiology 91(), 48-55. https://dx.doi.org/10.1016/j.jinsphys.2016.06.009). We had already cited Simmonds et al. (reference 214) but have added a couple of sentences about DlRhV.
To my knowledge the only species with a clear description of VLPs produced in venom gland isM. pulchricornis. The presence of particles or vesicles may be looking like virus in venom from some species has been reported but not confirmed by other mean than microscopy. This is covered in our text.
281-283 necessary since VLPs are produced in the ovary in this species? Indeed and we make this abundantly clear. The sentence is there to emphasise the variability (evolutionary plasticity) of VLPs even among closely-related species. We have not deleted it.
334-341 I don't understand the rationale of this paragraph. This paragraph emphasises further the complexity of studying host paralysis and paralysing venoms in parasitoid wasps since even wasp larval saliva can be paralytic and propose a question worthy of further investigation (appropriate to this review topic).
L344 what's mean changes in host biochemical composition ? It means changes in protein, lipid, carbohydrate and other host components. It seemed clear but we have modified it a little to be more explicit.
L346 the physical appearance of the hemolymph ? Yes, as stated
L362-364 this may help the teratocytes growth or to decrease the host defense thus indirectly benefits to the wasp larva. No, This wasp is an ectoparsitoid and does not produce teratocytes.
L367 Proteins up-regulated and down-regulated ? Thanks, now corrected.
L374, castration is mostly viewed as a competition for ressources between the parasitoid larva and the host reproduction. This is how we portay it
L376-380 the publications are mainly demonstrating that GGT is the enzyme responsible for castration. We have slighly modified the paragraph accordingly
Part 3.3 there is much more information on venom effects on the host immune system than the few reported here. We cover all cases that we are aware of that do not involved wasps with polydnaviruses.
Part 4; It seems that from here we start a new review on venom? From this point onwards we go through studies of particular venoms case by case and in more detail. What preceeded this was an overview of general venom roles.This seems perfectly appropriate to us and the other referees.
This is too long before starting the true review on the subject. We disagree because the following goes into far more detail on specific cases rather than giving an overview.
The text is very long and reports mainly what is in the publication. Yes, this is the purpose of a review. A synthesis effort should be made, the reader is already lost by the previous parts and don't really understand what you want to talk about. What synthesis is currently possible is given in the Conclusions section. I suggest this review to be shorten since a large part is not directly related to the subject. The purpose of this (and most reviews) is to provide a novice reader with a structured access to revevant literature and their summarised findings.
Please when you modify a figure from a publication even under CC-BY licence you should indicate it. DONE.
Overall although the authors have made some bibliographic efforts a large part of the manuscript is not clear and not focused on the subject. Disagree.
Best regards,
Donald & Buntika